# Nonparametric-based estimation method for river cross-sections with point cloud data from UAV photography

# URiver-X version 1.0 -methodology development

Keywords: Nonparametric, UAV, Regression, Point Cloud, River, Cross Section

Taesam Lee[1] and Kiyoung Sung[1]

[1] Department of Civil Engineering, ERI, Gyeongsang National University,
501 Jinju-daero, Jinju, Gyeongnam, South Korea, 660-701

Corresponding Author :

Taesam Lee, Ph.D.
Gyeongsang National University, Dept. of Civil Engineering
Tel)+82-55-772-1797, Fax)+82-55-772-1799
Email) tae3lee@gnu.ac.kr

# Abstract

Aerial surveying with unmanned aerial vehicles (UAVs) has been popularly employed in river management and flood monitoring. One of the major processes in UAV aerial surveying for river applications is to demarcate the cross-section of a river. From the photo images of aerial surveying, a point cloud dataset can be abstracted with the structure from motion (SfM) technique. To accurately demarcate the cross-section from the cloud points, an appropriate delineation technique is required to reproduce the characteristics of natural and manmade channels, including abrupt changes, bumps, and lined shapes, even though the basic shape of natural and manmade channels is a trapezoidal shape. Therefore, a nonparametric-based estimation technique, called the K-nearest neighbor local linear regression (KLR) model, was tested in the current study to demarcate the cross-section of a river with a point cloud dataset from aerial surveying. The proposed technique was tested with a simulated dataset based on trapezoidal channels and compared with the traditional polynomial regression model and another nonparametric technique, locally weighted scatterplot smoothing (LOWESS). Furthermore, the KLR model was applied to a real case study in the Migok-cheon stream, South Korea. The results indicate that the proposed KLR model can be a suitable alternative for demarcating the cross-section of a river with point cloud data from UAV aerial surveying by reproducing the critical characteristics of natural and manmade channels, including abrupt changes and small bumps, as well as the overall trapezoidal shape.

# 1. Introduction

Unmanned aerial vehicles (UAVs) have been popularly employed in recent years, especially to investigate and survey earth systems such as agriculture and coastal areas (Hugenholtz et al., 2013; Lin et al., 2018; Marfai et al., 2019; Remondino et al., 2011; Siebert and Teizer, 2014; Srivastava et al., 2020; Taddia et al., 2021; Wang et al., 2019; Watanabe and Kawahara, 2016; Yan et al., 2021). Furthermore, river management and fluvial networks have received critical attention for UAV applications (Gracchi et al., 2021; Langhammer, 2019; Lee et al., 2019; Sanhueza et al., 2019; Tomsett and Leyland, 2019). Additionally, flood monitoring and assessment are one of the major fields in which UAV aerial surveying data have been used (Anders et al., 2020; Andreadakis et al., 2020; Izumida et al., 2017; Kaewwilai, 2019; Perks et al., 2016; Zakaria et al., 2018).

For example, Andreadakis et al. (2020) employed a combination of SfM and optical granulometric techniques in estimating peak discharge and illustrated that the combined UAV technique can accurately determine peak discharge. Anders et al. (2020) tested different flying altitudes and area coverage orientations in semiarid and medium-relief areas with respect to cell size and vertical and horizontal accuracy. Perks et al. (2016) applied a novel algorithm to track features associated with free-surface velocity and to allow accurate geometric correction of velocity vectors. Those applications of UAV photographs to measure and analyze floods are based on the SfM technique.

The structure for a motion technique produces 3D information from overlapping images, where the structure refers to the relative parameters of aerial surveying, such as camera positions and focal lengths, and the relative positions of the corresponding features, while the motion refers to the movement of the camera (Javernick et al., 2014; Marteau et al., 2017; Smith et al., 2014). A

dense point cloud can be determined from the SfM. These point clouds are converted from an arbitrary coordinate system to a geographical coordinate system with camera position and focal length information or by associating reference points on the ground, called ground control points (GCPs), with known coordinates. A point cloud is a set of 3-dimensional points located in space. The 3D locations of a point cloud can be determined from a sensor by emitting pulses and calculating them with the position of the sensor and the pulse direction. Here, the sensor refers to a photogrammetry camera in the current study.

In UAV aerial surveying applications for river management and flood analysis, the demarcation of the cross-section of a river is the critical process. Accurate demarcation of the cross-section is mostly required to calculate the peak discharge and flow amount. However, the dense cloud point dataset obtained from UAV aerial surveying and the SfM technique mostly contains errors and does not provide direct cross-sectional information. An appropriate technique to demarcate the cross-section from the point cloud dataset is necessary to develop.

The demarcation of the cross-section in a river has been mostly made with a digital elevation model (DEM) in the literature (Gichamo et al., 2012; Petikas et al., 2020a, b; Pilotti, 2016). For example, Petikas et al. (2020b) proposed a novel method to automatically extract river cross-sections from a DEM along with a parametric cross-section extraction algorithm. However, a cross-sectional algorithm for the cloud point dataset of UAV aerial surveying has not been tested in depth since the characteristics of the point cloud dataset are far different from the DEM in that a study area for UAV aerial surveying is commonly smaller and many more points can be acquired from UAV aerial surveying.

Therefore, the current study proposes a demarcation technique for river cross-sections from the point clouds of UAV aerial surveying. A cross-section of natural rivers contains abrupt changes and small bumps as well as smooth variations even though it normally has a trapezoidal shape. The demarcation technique must reproduce the characteristics of natural rivers as well as the abrupt changes in a manmade channel. The proposed demarcation model was tested to determine whether to reproduce the characteristics of natural river and manmade channels.

## 2. Mathematical Description

With the point cloud data obtained from UAV aerial surveying and postprocessing, the river cross-section must be demarcated. Polynomial regression can be simply applied to the point data. However, the fixed shape of the polynomial regression along with its order is limited to the highly varied shape of the cross-section. Therefore, a nonparametric regression approach is adopted in the current study, especially K-nearest neighbor local regression. A detailed description of polynomial regression and the nonparametric regression model is shown in the following.

### 2.1. Polynomial Regression

A polynomial regression model can be used when the relationship between a predictor ($x$) and an explanatory variable (y) is nonlinear or curvilinear. The $k^{th}$-order polynomial regression can be expressed as

$$y = \beta_0 + \beta_1 x + \beta_2 x^2 + \cdots + \beta_k x^k + \epsilon = \sum_{i=0}^{k} \beta_i x^i + \epsilon = \mathbf{x}\boldsymbol{\beta} + \epsilon \qquad (1)$$

where $\epsilon$ is considered to be random noise with zero mean.

## 2.2. KNN-based Local Linear Regression (KLR)

Assume that the current condition of the predictors $x_t$ with the observed data pairs $(x_i, y_i)$, for $i$ = 1,…,$n$, is given for the $n$ number of data points (i.e., the selected cloud points). The number of neighbors ($k$) is also assumed to be known. The predictor $Y_t$ is estimated according to the following steps:

(a) Estimate the distances between the current and observed states of the predictors for all $n$ observations, as follows:

$$D_j = \left(x_j - x_t\right)^2 \qquad j=1\ldots,n \quad (2)$$

(b) Store the location indices for the $k$ smallest distances.

(c) Fit the local linear regression to the observed dataset of the selected location indices [$x_{(p)}$, $y_{(p)}$] for $p$ = 1,…,$k$, where ($p$) indicates the $p^{th}$ decreasing ordered location index relative to the distance measure in step (a).

(c-1) Build the weight matrix using the simple selection weight as follows:

$$\boldsymbol{W}_{KLR} = diag\left[\frac{1}{\delta}, \frac{1/2}{\delta}, \ldots, \frac{1/k}{\delta}\right] \quad (3)$$

where $\delta = \sum_{p=1}^{k} 1/p$.

$$\overleftrightarrow{\mathbf{X}}_t = \begin{pmatrix} 1 & x_t - x_{(1)} \\ 1 & x_t - x_{(2)} \\ 1 & \vdots \\ 1 & x_t - x_{(k)} \end{pmatrix} \quad (4)$$

(c-3) Estimate the parameter vector $\hat{\boldsymbol{\beta}}_t^{KLR}$ with the weighted least square estimator from the weight matrix $\mathbf{W}_{KLR}$ in Eq. (3) as

$$\hat{\boldsymbol{\beta}}_t^{KLR} = (\overleftrightarrow{\mathbf{X}}_t^{T}\boldsymbol{W}_{KLR}\overleftrightarrow{\mathbf{X}}_t)^{-1}\overleftrightarrow{\mathbf{X}}_t^{T}\boldsymbol{W}_{KLR}\mathbf{y}_{KLR} \tag{5}$$

where $\mathbf{y}_{KLR}$ is the corresponding predicted value for the ordered observations $[y_{(1)}, y_{(2)}, \ldots, y_{(k)}]^T$.

(d) Estimate the current predictor as follows:

$$y_t = \vec{\mathbf{x}}_t^T \hat{\boldsymbol{\beta}}_t^{KLR} \tag{6}$$

where $\vec{\mathbf{x}}_t = (1 \;\; x_t)$.

(e) Repeat steps (a)-(d) until the required data are simulated.

For selecting the number of neighbors *k*, a heuristic approach for estimating *k* for the KNNR model is given by $k = \sqrt{n}$ (Lall and Sharma, 1996; Lee and Ouarda, 2011; Lee et al., 2010). Therefore, Lee et al. (2017) suggested a heuristic approach for KLR in which they suggested that the multiplier be

$$k = a\sqrt{n} \tag{7}$$

where a is a multiplier and is a positive integer (i.e., 1,2,3,4).

As noted, only the partial dataset is employed for the observations rather than the whole observation dataset, unlike other regressions. For the point cloud dataset from UAV photography, this proposed approach in the current study is highly advantageous since the neighboring data point

is sufficient and the fitting of the target point must not be affected by the points that are far away from the target point. This advantage is further discussed in the results section.

# 3. Simulation Study

The performance of the KLR model in fitting the point cloud data for river cross-sections is tested with the simulated point cloud data.

## 3.1. Model Description and Fitting

A river cross-section is generally trapezoidal due to maximum discharge and easy construction (Chow, 1959). Therefore, a trapezoidal channel was assumed with a 4 m top at both sides and a 6 m base width as well as a 1:1 side slope with a 6 m height, as shown by the thick solid blue line of Figure 1. The channel points were assumed to be measured with 0.1 m intervals, for a total of 161 points. It is assumed that these points work as cloud points that UAV cameras might capture in aerial surveying. The assumed cloud point dataset was generated based on the assumed 161 points (see the thick solid blue line in Figure 2), as follows:

$$Z = Y + \varepsilon \quad (8)$$

where $Y$ is the assumed points, and $\varepsilon \sim N(0, \sigma_\varepsilon^2)$, i.e., normally distributed error. Note that the generated data (Z) are presented with red circles in Figure 1.

In the current study, $\sigma_\varepsilon^2$=0.2 was used following similar variability of observed data after testing several values. The magnitude of this error variance ($\sigma_\varepsilon^2$) represents the differences in the photo locations for the same cloud point of the real ground location (i.e., $Y$ in this case). High variance indicates that extracted point clouds include high errors, and vice versa.

In Figure 1 and Figure 2, the simulated data are presented with red circles. The number of simulated data points was chosen to be 2 times and 10 times the assumed 161 points that were applied for the assumable measured trapezoid line (i.e., 322 and 1610 points), as shown in Figure 1 and Figure 2, respectively. Note that the recommended overlap is 70–80% frontal and 60% side in general cases. In this overlapping case, each cross-section point might be captured approximately 10 times. Therefore, the number of simulated data points is set to 10 times the number of trapezoidal channel points (a total of 1610), as shown in Figure 2. Additionally, there are some portions in which overlapping might not be achieved. Minimal overlap to be a point cloud is at least 2 times, and 2 times the channel points were also tested, as shown in Figure 1.

### 3.2. Simulation Results

In Figure 1, the fitted cross-section line to the KLR model is shown with the dashed black line for the generated data case with 2 times the assumed target points, while the simulated data are presented with the red circles as noted. Note that the multiplier ($a$) for the number of nearest neighbors as in $k= a\sqrt{n}$ in Eq. (7) was tested in this figure. As shown in Figure 1, the multiplier a= 1, 2, 3, 4 is shown in each panel. The fitted KLR line with a smaller multiplier presents more irregularity, while the line with a higher multiplier appears to be smooth. For example, the top part of the trapezoid channel with the 22 m of the y-coordinate shows that the KLR line with the multiplier $a$=1 (the panel($a$) of Figure 1) was drawn rather coarse, but the straight shape of the original channel is preserved. At the same time, the fitted line with the high multiplier $a$=4 (Panel ($d$) of Figure 1) presents a very smooth feature and presents the original top and bottom horizontal parts, which are rather too curved.

The multipliers of $a$=2 and 3 in the fitted KLR model, as shown (Panels (*c*) and (*d*) of Figure 1), appear to mix the smooth and horizontal features well by fitting the top and bottom horizontal lines, and the angled part of the original channel is reproduced well. This finding indicates that an appropriate multiplier ($a$ in Eq. (7)) is required to present the straight and angled trapezoid channel better.

This characteristic continues to the case with the high number of captured cloud points, as shown in Figure 2 (i.e., 10 times the target points, for a total of 1610 points, as shown with the red circles in this figure). It is comparable to the case of 2 times the target points in Figure 1 in that all of the fitted line with the KLR model with the case of 10 times presents better the original trapezoid channel than the case of 2 times. It is obvious that a higher number of points can significantly improve the quality of the KLR model since the nonparametric KLR model directly applies the observed data and its performance highly depends on the number of data points. In other words, while parametric models such as linear regression and polynomial regression estimate the parameters from the data and the parameters are employed, the nonparametric KLR model employs the data itself directly to estimate the cross-section. It can be appreciated that UAV aerial photography usually captures a large enough number of points to produce overlapping points as many as 10 times the target points.

The horizontal and angled trapezoid shape (i.e., the solid thick blue line in Figure 2) is reproduced well by the KLR model (see the dashed black line), even though a coarse zig-zag line is still observed in the case of the small multiplier (i.e., $a$=1, see Panel (*a*) of Figure 2), and the angled portion is too curved in the case of the high multiplier (i.e., $a$=4, see Panel (*d*) of Figure 2).

The results of Figure 1 and Figure 2 illustrate that a value between 2 and 3 can be a good selection for the multiplier. Further testing was performed to select the multiplier for the number of nearest neighbors by varying the multiplier from 0.5 to 5.0 with a 0.5 interval. The root mean square error was estimated as

$$RMSE = \sqrt{\frac{1}{\mathrm{N}}\sum_{t=1}^{N}(y_t^{KLR} - Y_t)^2} \quad (9)$$

where $y_t^{KLR}$ is the KLR estimate from Eq. (7), and $Y_t$ represents the original points with N trapezoid points (here, 161 points).

The RMSE results of the KLR estimate with different multipliers (i.e., *a* in Eq. (7)) are shown in Figure 3 for the case of 2 times (top panel) the original trapezoidal points and the case of 10 times (bottom panel). In the 2 and 10 times cases, the optimum multiplier (i.e., the smallest multiplier *a*) can be selected to be between 1.5 and 2.5. To fully reveal the characteristics of the multiplier with multiple simulations, all of the multiple simulations from 1 to 12, indicating the number of overlapped photos, were tested while finding the optimum multiplier. The result in Figure 4 shows that a smaller optimum multiplier is selected with a smaller number of overlapped photos (or multiple simulation points) as much as 1.5–2.0, and vice versa as much as 2.0–2.5. Since the number of overlapping photos might be difficult to know and each point does not have the same number of points in real UAV aerial survey, the multiplier is suggested to be 2.0 in the current study.

To compare other approaches to fit the point cloud in demarcating the cross-section of a river, polynomial regression and locally weighted scatterplot smoothing (LOWESS) were also tested. The result is presented in the top and bottom panels of Figure 5 and Figure 6 for the fitted line to
Open Access EGU

the polynomial regression (top panel) and the LOWESS model, respectively. The fitted line to the polynomial regression of $2^{nd}$ degree (see the thick dash-dotted yellow line with the circle marker in the top panel of Figure 5 and Figure 6) does not reproduce the top and bottom horizontal lines of the trapezoid channel well. Better performance in the $4^{th}$ degree polynomial regression model is presented (see the dotted b line with the reverse triangle marker). However, the depth of the trapezoid center is overestimated. Other degrees of polynomial regression models were also tested, but no better performance was observed.

Furthermore, the LOWESS model was additionally fitted to the simulated trapezoid channel data. Note that the LOWESS model is also a nonparametric regression model, and its detailed description is presented in the Appendix. The major difference between the LOWESS model and the KLR model is that the LOWESS model includes all of the observed data in the estimate, as shown in Eqs. (A.3) and (A.4), while the KLR model includes only the k-nearest neighbor observations, as in Eq. (4). The performance of the LOWESS and KLR models was compared in detail in Lee et al. (2017) for the heteroscedastic relation of time series data. The result in the study of Lee et al. (2017) indicated that the KLR model reproduces an abrupt change in the heteroscedastic relation. The results of the LOWESS model are presented in the bottom panels of Figure 5 and Figure 6. The results indicate that the bottom part of the trapezoidal channel is reproduced well with the LOWESS model. However, the model does not reproduce the abruptly curved area well.

Further nonparametric models such as LOWESS and other regression models such as logistic regression (Ahmad et al., 1988; Elek and Márkus, 2004; Orlowsky et al., 2010; Simonoff, 1996) can be tested. However, the simulation study with the trapezoid channel that is similar to the real river cross-section shows that the presented KLR nonparametric model is suitable for demarcating

the cross-section of a river. The major reason for the good performance is that the KLR model employs only k-nearest neighbor observations. This approach might not be beneficial when an overall trend is needed and not enough observations are available. However, the point cloud data taken from UAV aerial surveying often provide a large enough number of points in the data set. Furthermore, the shape of the cross-sections in a natural river is irregular, and abrupt changes can be easily observed. This feature can be captured only through fitting nearby observations. Therefore, the KLR model might be a suitable alternative to demarcating the cross-section of a river with the cloud point dataset.

# 4. Case Study

## 4.1. Study Area and Data Acquisition

### *4.1.1. Study Area*

The study area is located in the Migok-cheon stream flowing through Hapcheon-gun, South Korea, as shown in Figure 7. The Migok-cheon stream has an 8.8 km length and 13.9 $km^2$ watershed area. The slope of the stream is approximately 1/50~1/400, and the study area has a slope of 1/350. This stream conjuncts to the Hwanggang River at the end of the stream, and the Hwanggang River is joined into the Nakdong River directly afterward; the Nakdong River is one of the four largest rivers in South Korea. Therefore, the Migok-cheon stream is highly affected by the water levels of the Hwanggang River and Nakdong River.

In the middle of the Hwanggang River, the Hapcheon dam is located for electric generation and water resources. The upstream Hapcheon River consists of a number of mountains, and the slope is high, producing rapid floods and short concentration times to induce floods. For example,

in August 2020, the Hapcheon dam outflowed a large amount of water downstream and induced a high water level in the Hwanggang River. A number of streams joining the Hwanggang River overflowed due to the high level of the Hwanggang River, including the Migok-cheon stream (Seong et al., 2020). To reduce damage from floods in the area of the Migok-cheon stream, an early warning system for floods is being considered. For the early-warning system for floods, detailed cross-sections of the Migok-cheon stream must be requested to decide which water level is appropriate for an alarm.

#### *4.1.2. Data Acquisition*

##### **Specification of Employed UAV**

Aerial photos over the selected Migok-cheon were obtained with the unmanned aerial vehicle (also termed drone), DJI Phantom 4. This UAV is one of the most popular professional drones on the market and contains an advanced stereo vision positioning system that provides precise hovering even without satellite positioning support (Hamdi et al., 2019). The camera applied is FC3411 with ISO-110, and the images taken from DJI Phantom 4 are 5472x3648 pixels at approximately 10 M. Pix4Dcapture was employed to map the target area.

##### **Ground Control Points**

Ground control points (GCPs) are the points on the ground that have measured or known coordinates. To obtain GCPs, 10 specific points were measured over the target area on the ground with global positioning system (GPS) surveying. The EMLID Reach RS2 (https://emlid.com/reachrs2/), multiband RTK GNSS receiver, with centimeter precision, was employed for GPS surveying.

### Data Processing (WebODM)

The aerial photos were postprocessed to build a point cloud dataset with WebODM. The WebODM (https://github.com/OpenDroneMap/WebODM) is an open source tool for generating map point clouds, terrain and 3D surface models from aerial images.

## 4.2. Distance Measurement of the Point Cloud

The point cloud data from UAV photography are presented with Transverse Mercator (TM) projection for x, y, and z. The TM projection is a conformal projection presented by Lambert in 1772. To demarcate a cross section of a river, the point cloud data must be projected to a new coordinate system.

As an example in Figure 8, the new coordinate system can be based on the line that connects N and L points presented with the thick red line in Panel (a). The extended thick red line is designated as the new x-coordinate, as shown in Panel (b), and the same z-axis can be defined as the original TM data. The y-coordinate can be chosen as the axis that is perpendicular to the x-coordinate. Let us assume that point M, as in Panel (b), is selected among the selected point clouds contained in the NL line. Note that the thick red line in Panel (a) is a group of selected points from the point cloud data for defining the cross-section of the river, as shown in Panel (b).

All of the selected red points must be aligned according to the distance from the datum point (here, N) with the new coordinate system. The new distance for the new x-coordinate can be defined as *k*, as shown in Panel (c). This distance is estimated with the following equations.

The distances of *l*, *m*, and *n* with the TM coordinate can be estimated as follows. For example, $N_{TM}(x)$ represents the x-coordinate of the TM projection for point *N*.

$$l = \sqrt{\left(N_{TM}(x) - M_{TM}(x)\right)^2 + \left(N_{TM}(y) - M_{TM}(y)\right)^2} \tag{10}$$

$$m = \sqrt{\left(N_{TM}(x) - L_{TM}(x)\right)^2 + \left(N_{TM}(y) - L_{TM}(y)\right)^2} \tag{11}$$

$$n = \sqrt{\left(L_{TM}(x) - M_{TM}(x)\right)^2 + \left(L_{TM}(y) - M_{TM}(y)\right)^2} \tag{12}$$

From the calculated angle of MNL ($\Theta$) in Eq. (13), the new x-coordinate distance ($k$) can be calculated as in Eq. (14) with the law of cosines (i.e., $n^2 = l^2 + m^2 - 2lmcos\theta$) as the following:

$$cos\theta = \frac{l^2+m^2-n^2}{2lm} \tag{13}$$

$$k = l\ cos\theta \tag{14}$$

## 4.3. Results

### *4.3.1. Selected sites for cross-section*

The two tested sites in the Migok-cheon stream are presented in Figure 9. The overall produced point cloud dataset for the UAV surveying area is presented in the left panel of Figure 9, and the picture of the left panel consists of only the collected points. Site-1 is located in the middle of the study area, while Site-2 is in the upper part of the area. Since the nearby area of Site-1 is located in the middle of the UAV surveying coverage, several images can be overleaped and captured for the same points.

Therefore, the number of points for demarcating a cross-section of the river might be sufficient to capture the detailed characteristics of the cross-section (see the top-right panel of Figure 9). In contrast, Site-2 is located at the upper part of the coverage area, and overlapping

images might be limited, which indicates that the number of points to capture a target cross-section is also limited. Furthermore, a part of the cross-sectional area can be missing due to technical and environmental limitations such as waterbodies and insufficient overlapping images. For example, there are some areas in which no cloud point data exist, as on the right side of Site -1. This point is intentionally selected to verify the model performance in such a case.

#### *4.3.2. Demarcation of the selected cross-sections*

The demarcated cross-sections for the selected sites (i.e., Site-1 and Site 2) are presented in Figure 10 and Figure 11, respectively. In Figure 10, the extracted points for Site-1 are presented with red circles. As noted, a number of points are extracted from the UAV aerial photographs for Site-1 since the site is located in the middle of the coverage. The KLR fitted line shown with the dashed yellow line indicates that the fitted line reproduces the characteristics of the natural cross-section of the river well, including the overall trapezoidal shape and the natural bumps at the bottom. This line is compared with the field measurement reported in Donggwang Engr. (2004). Slight differences can be seen between the field measurement (shown with the solid blue line with the x marker) and the KLR fitted line since the field measurement took place approximately 17 years ago. However, the overall characteristics match well with each other, which indicates that the KLR fitted line fairly demarcates the cross-section of the river.

The cross-section of Site-2 is presented in Figure 11 and shows that the middle part of the river has no cloud point data. The KLR fitted line shows that the overall characteristics of the cross-section are reproduced. Even the missing part of the cross-section is also interpolated well with the KLR model by comparing the field measurements (see the solid blue line with the x marker in Figure 11). Some differences between the fitted line and the field measurement might result from the year difference. The result indicates that the KLR method can reproduce the

characteristics of the cross-section of a natural river. Furthermore, the missing part of the aerial surveying can be filled up with the interpolation of the KLR method.

### *4.3.3. Estimating cross-sectional area and wetted perimeter*

One of the main reasons to delineate the cross-section of a river is to model and estimate the flow in the cross-section. The area and perimeter are essential to estimate for modeling the discharge of the cross-section. With the KLR fitted line for the cross-sections of the river shown in Figure 10 and Figure 11 for Site-1 and Site-2, respectively, it is straightforward to estimate the area and the perimeter. The area and perimeter can be estimated with the fixed height (H) as

$$A(H) = \sum_{t\in[(H-y_t)>0]} \frac{[(H-y_t)+(H-y_{t+1})]}{2} \Delta x \qquad (15)$$

$$P(H) = \sum_{t\in[(H-y_t)>0]} \sqrt{(y_{t+1}-y_t)^2 + \Delta x^2} \qquad (16)$$

where $y_t$ is the fitted KLR line as in Eq. (6), and $\Delta x$ is the interval of the x-coordinate in estimating $y_t$.

The estimated $A(H)$ and $P(H)$ are presented in Figure 12 and Figure 13 for Site-1 and Site-2, respectively. The area of Site-1 is exponentially increased, while the perimeter is increased at different steps according to the heights, as shown in Figure 12. As seen in Figure 10, the width of the cross-section increases as the height increases, and this feature affects the exponential increase in the area as the height increases. At heights between 17.0 m and 18.5 m, the perimeter increases rapidly as the height increases, as shown in the lower panel of Figure 12, since the shape of the cross-section is rather flat in this range of heights, as shown in Figure 10. The other part of the perimeter increases linearly except for a higher slope between 21 and 21.5 since the width becomes

wider in this range of heights. Note that the area and perimeter outside the bank is excluded because it is between 0 and approximately 10 m and over 37 m in the x-coordinate of Figure 11.

Similar features of the area and perimeter along with the height to Site-1 can be observed at those of Site-2, as shown in Figure 13. This exponential and S-shaped increase in the area and perimeter is a typical characteristic at the cross-section of natural rivers and trapezoidal channels. The results of the area and perimeter show that the proposed KLR method can be a reasonable alternative in demarcating the cross-section of a river obtained from a point cloud dataset.

## 5. Summary and Conclusions

The current study presents a nonparametric fitting method, the KLR, to the point cloud data from UAV areal surveying to demarcate the cross-section of a river. Other than general fitting data, the cross-section of a natural river generally contains sudden variation, an angled shape, and even bumps as well as a linear shape even though the overall shape of a cross-section for a river is trapezoidal. To accommodate all of those features of the natural cross-section, a highly flexible fitting model is requested. Furthermore, the observed datae point is large enough for the point cloud dataset. Therefore, the KLR model was chosen to fit the point cloud data for the cross-section. The results conclude that the tested KLR model can reproduce the critical characteristics of the cross-section of natural rivers with the point cloud data from UAV aerial surveying.

The major limitation of the point cloud data employed in the current study is that RGB photographs were employed and the vegetation inside the river could generate an obscure cross-sectional shape. Further optical instruments, such as hyperspectral and lidar sensors, could be

tested to overcome this limitation. However, a perfect solution that can remove the vegetation inside rivers has not yet been developed. To avoid this issue, points of the cross-section where little vegetation exists can be selected.

This KLR method can be easily adopted for other demarcation cases, such as buildings and structures. The proposed KLR method is a rather simple and direct approach for demarcating an area and structures. Additionally, other nonparametric techniques, such as LOWESS, can be further tested with extensive testing and adjustment. In the current study, the KLR model alone was focused on since the clustered data setting is obvious and easy to apply.

## 6. Code and data availability

All the employed code and excel files are available at Mendeley Data in <http://dx.doi.org/10.17632/xdw4cgnvhm.1>.

**Competing interests**

The author declares that they have no conflict of interest.

**Author Contribution**

T.L. carried out the research plan and programming as well as supervising while K.S. performed writing and collection data.

**Acknowledgments**

This work was supported by a National Research Foundation of Korea (NRF) grant funded by the Korean Government (MEST).

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

## Appendix: LOcally WEighted Scatterplot Smoothing (LOWESS)

LOWESS was proposed by Cleveland (1979) as a nonparametric regression. The LOWESS with one explanatory variable and one predictor variable can be defined as

$$y_t = m(x_t) + \varepsilon_t \qquad \text{(A.1)}$$

where the regression curve $m(\mathbf{y}_t)$ is the conditional expectation $m(x_t) = E(Y|X = x_t)$. The LOWESS estimate can be defined as

$$\hat{m}_{LOWESS}(x_t) = \vec{x}_t^T \hat{\boldsymbol{\beta}}_t^{LOWESS} \qquad \text{(A.2)}$$

where

$$\hat{\boldsymbol{\beta}}_t^{LOWESS} = (\overleftrightarrow{\mathbf{X}}_t^{\,T} \boldsymbol{W}_t \overleftrightarrow{\mathbf{X}}_t)^{-1} \overleftrightarrow{\mathbf{X}}_t^{\,T} \boldsymbol{W}_t \mathbf{y} \qquad \text{(A.3)}$$

with

$$\overleftrightarrow{\mathbf{X}}_t = \begin{pmatrix} 1 & x_t^1 - x_1^1 \\ 1 & x_t^1 - x_2^1 \\ 1 & \vdots \\ 1 & x_t^1 - x_n^1 \end{pmatrix} \qquad \text{(A.4)}$$

and

$$\boldsymbol{W}_t = \boldsymbol{H}^{-1} diag[K_d(\boldsymbol{H}^{-1}(\boldsymbol{x}_t - \boldsymbol{x}_1)), \cdots, K(\boldsymbol{H}^{-1}(\boldsymbol{x}_t - \boldsymbol{x}_n))] \qquad \text{(A.5)}$$

with the bandwidth matrix, **H**. The major characteristic of LOWESS is to employ the following kernel function:

$$K_d(z) = \begin{cases} (1 - |z|^3)^3 & |z| < 1 \\ 0 & \text{otherwise} \end{cases} \tag{A.6}$$

# Figure

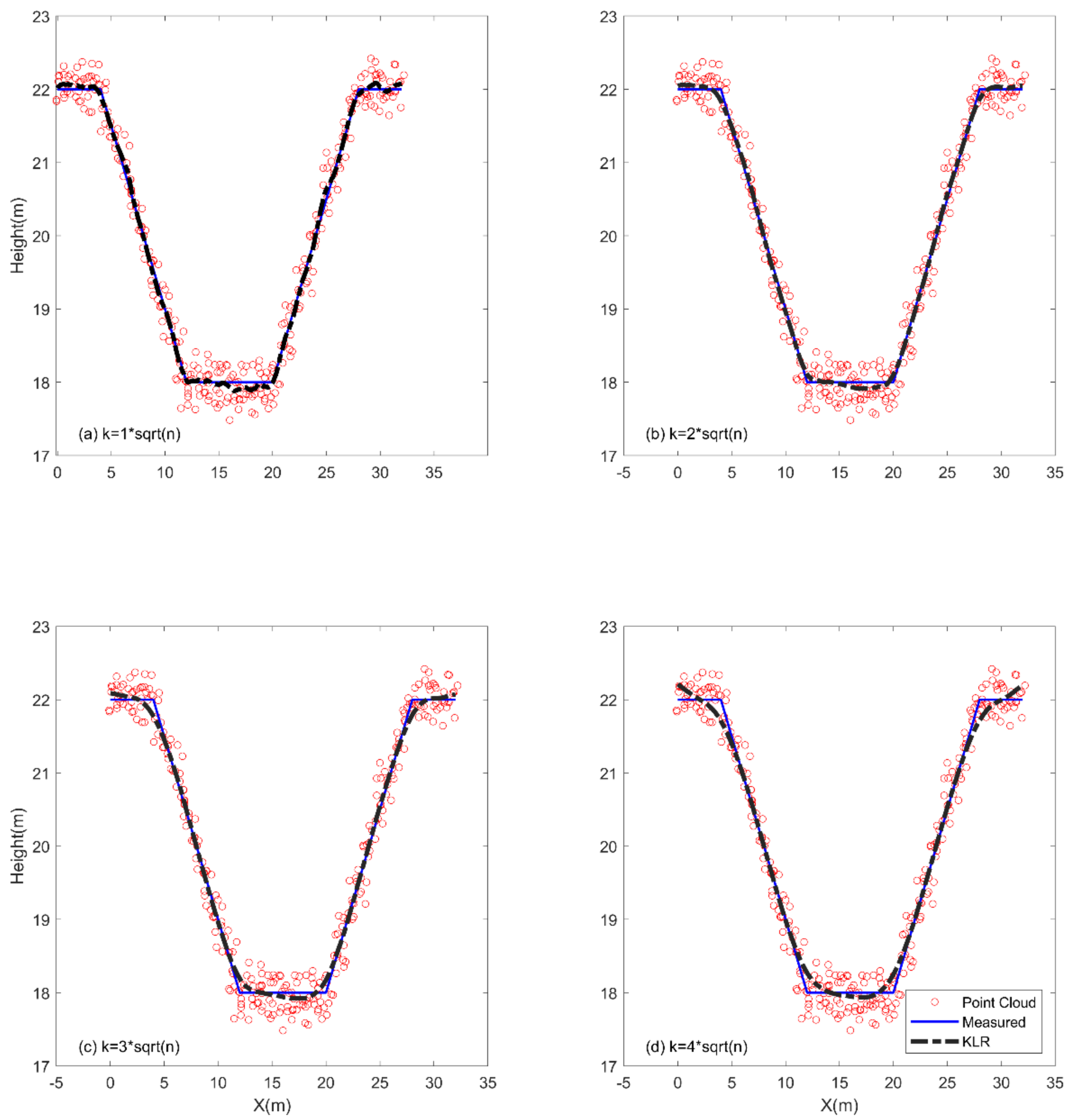

Figure 1. Assumed trapezoid channel to test the KLR model for point cloud data with different portions of the number of neighbors ($k = a\sqrt{n}$, here $a$=1, 2, 3, and 4 at each panel). Note that (1) the trapezoid sections are consistent with a 4 m top both sides and a 6 m base width as well as a 1:1 side slope with a 6 m height; (2) the number of points for the channel is divided at each 0.1 m, to total 161 points; (3) it is assumed that 10 times the divided data are collected, to total 322 points; and (4) the elevation of the bottom channel was assumed to be 18 m.

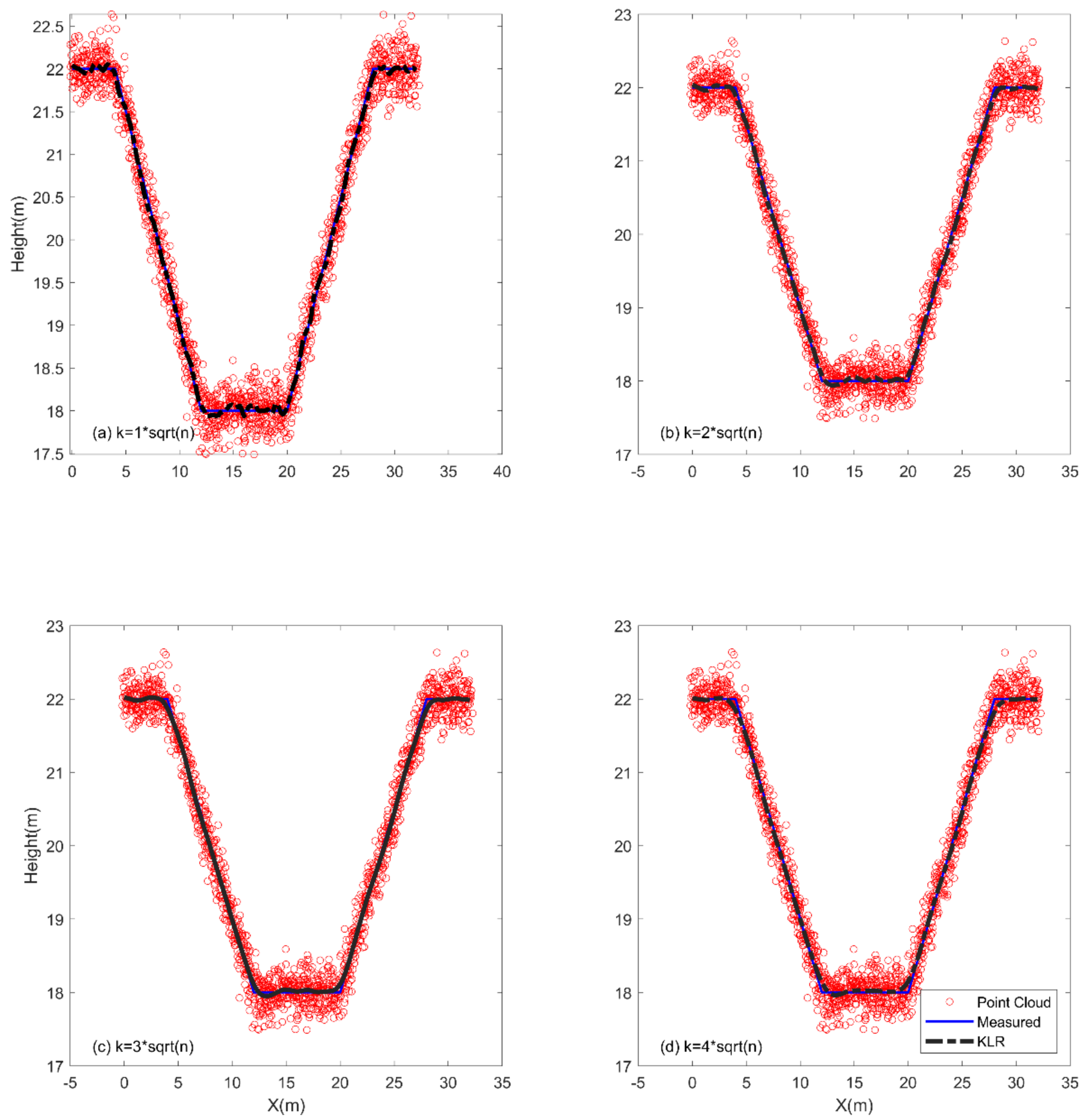

Figure 2. Assumed trapezoidal channel to test the KLR model for point cloud data with different multipliers of the number of neighbors ($k = a\sqrt{n}$). Note that (1) the trapezoidal sections are consistent with a 4 m top both sides and a 6 m base width as well as a 1:1 side slope with a 6 m height; (2) the number of points for the channel is divided at each 0.1 m, for a total of 161 points; and (3) it is assumed that 10 times the divided data are collected, for a total of 1610 points.

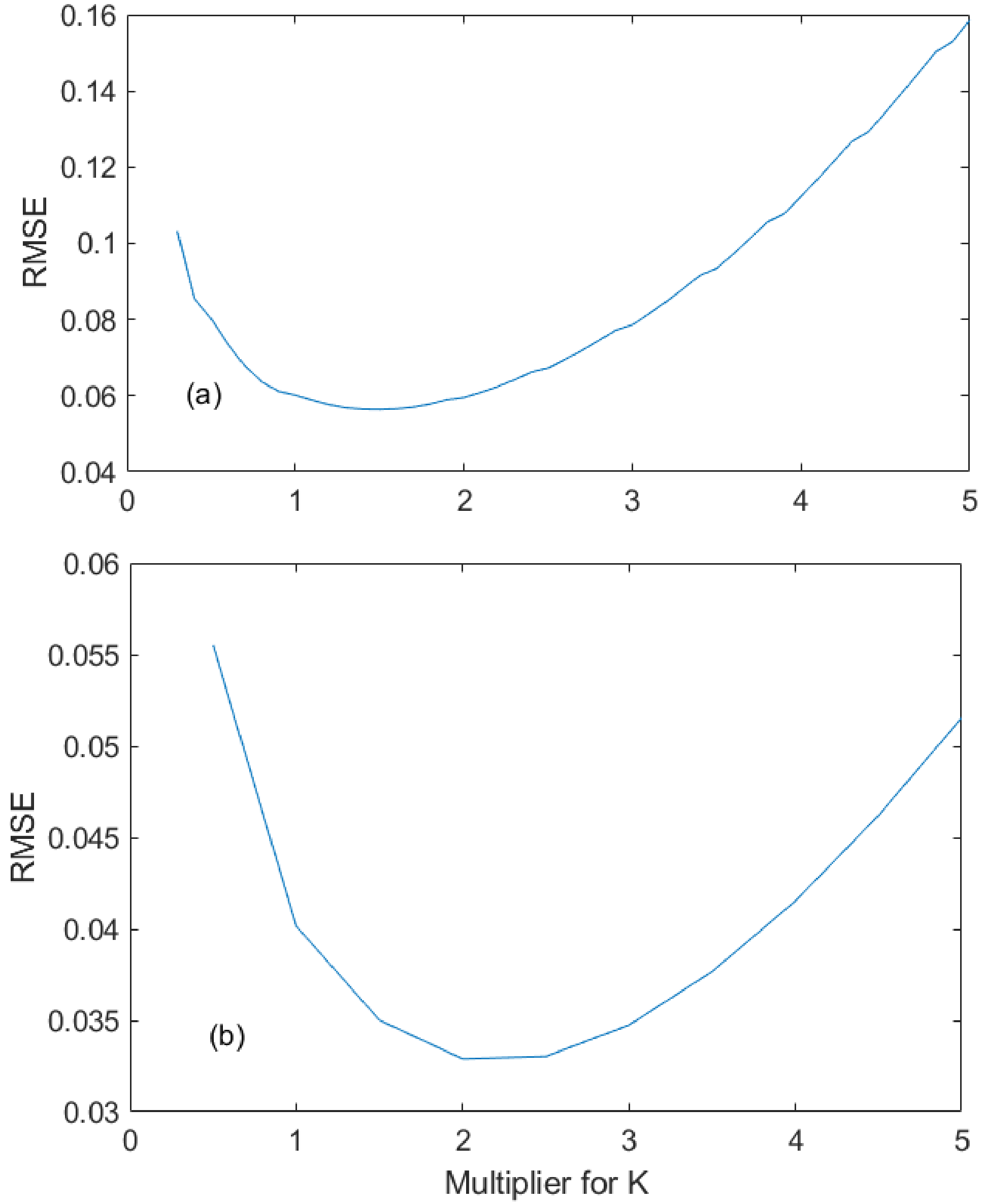

Figure 3. Root mean square error (RMSE) between the KLR estimate with different multipliers ($a$) of the number of neighbors ($k = a\sqrt{n}$) and the original trapezoid points for the case of 2 times the original points (Panel (a)) and 10 times (Panel (b)).

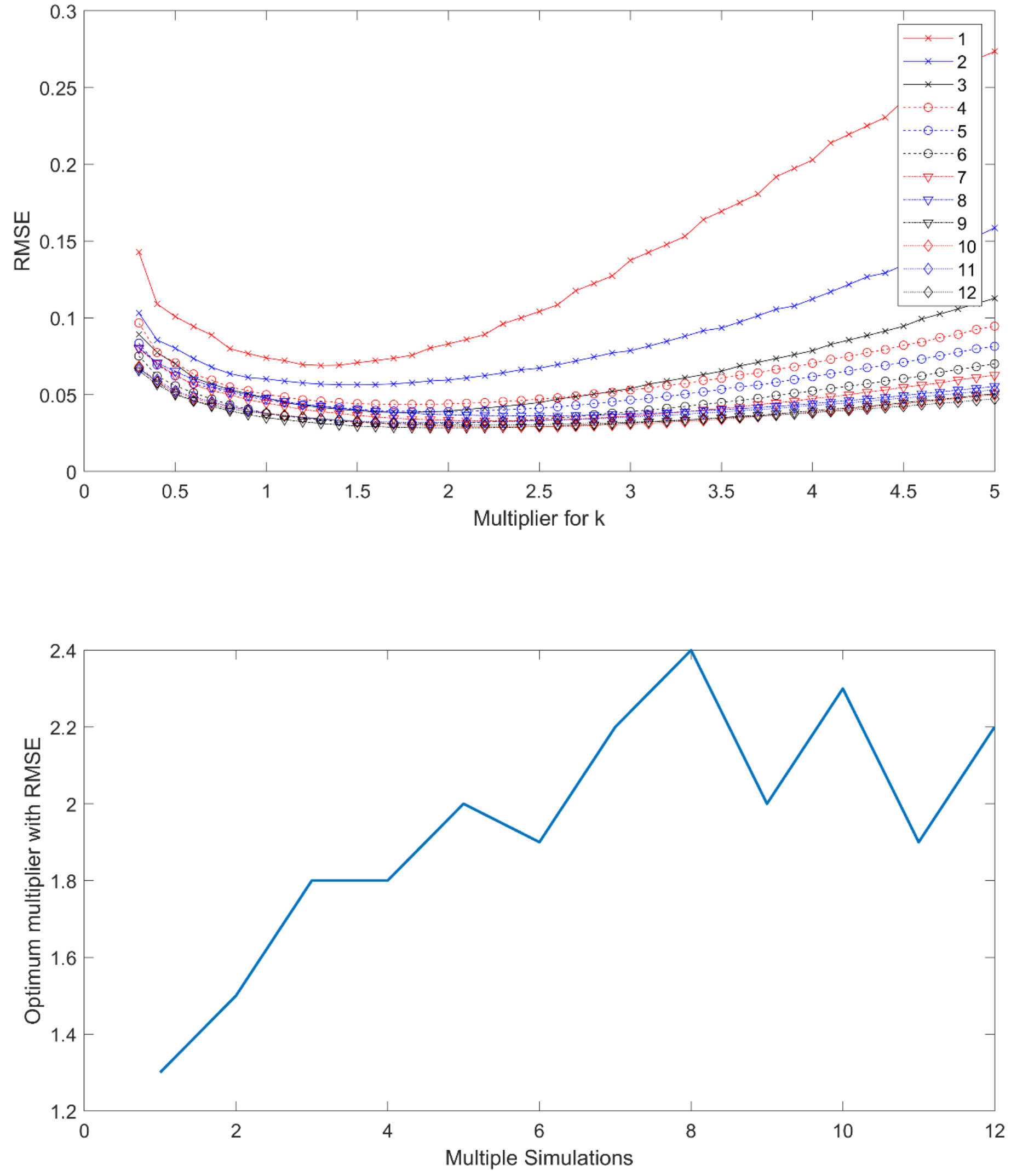

Figure 4. Root mean square error (RMSE) between the KLR estimate with different multipliers of for number of neighbors ($k = a\sqrt{n}$) and the original trapezoid points for all of the cases between 1 and 12 times the original points (top panel) as well as the optimum multiplier with the RMSE value at the top panel for each multiple simulation. Note that increasing the number of multiple simulations indicates that the number of overlapped photos is increased and the cloud points are multiplied.

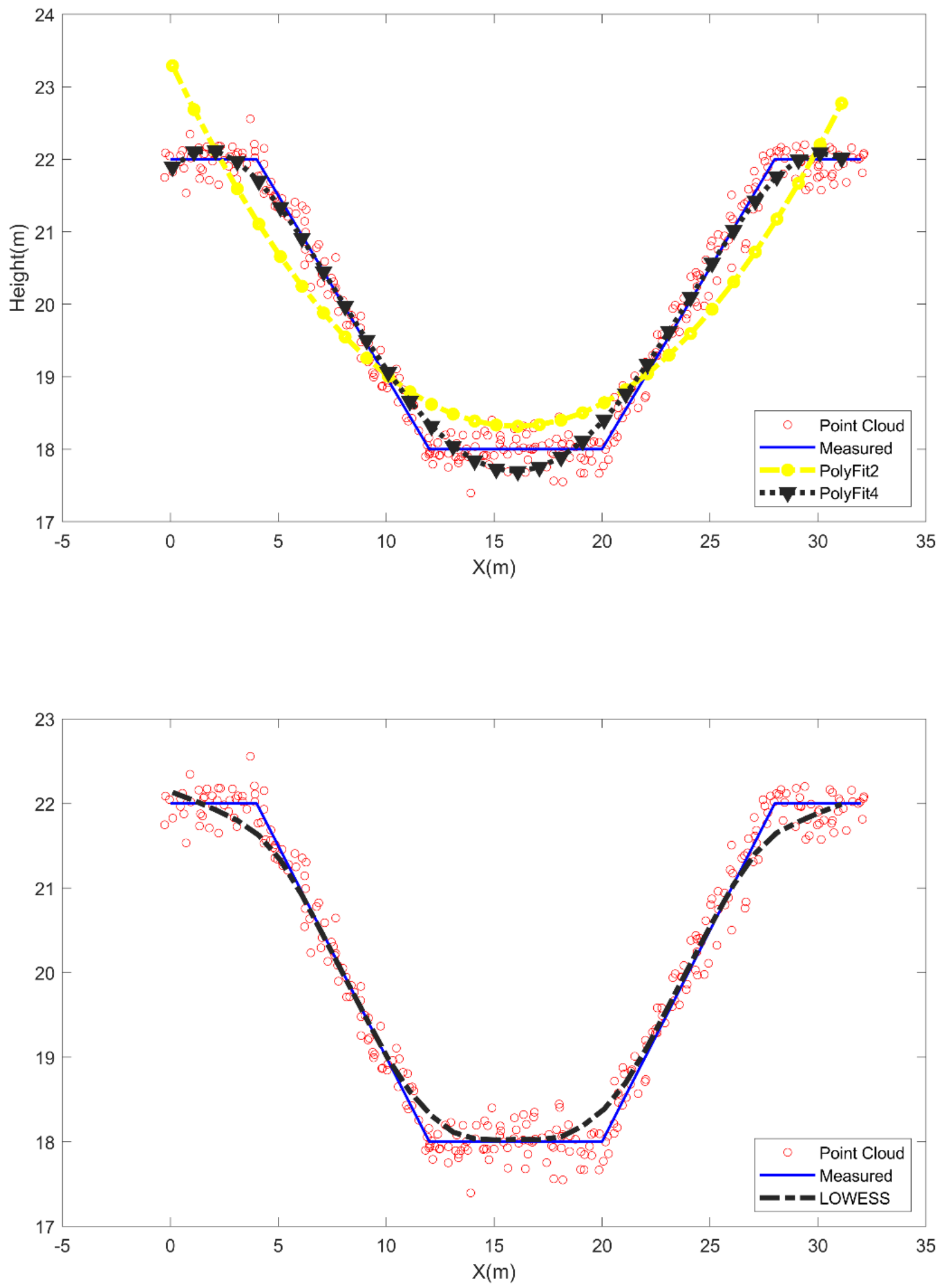

Figure 5. Polynomial regression (top panel) and LOWESS (bottom panel) were fitted to the simulated point cloud data of 2 times the assumed cloud point (red circles) and assumed trapezoidal channel (blue line).

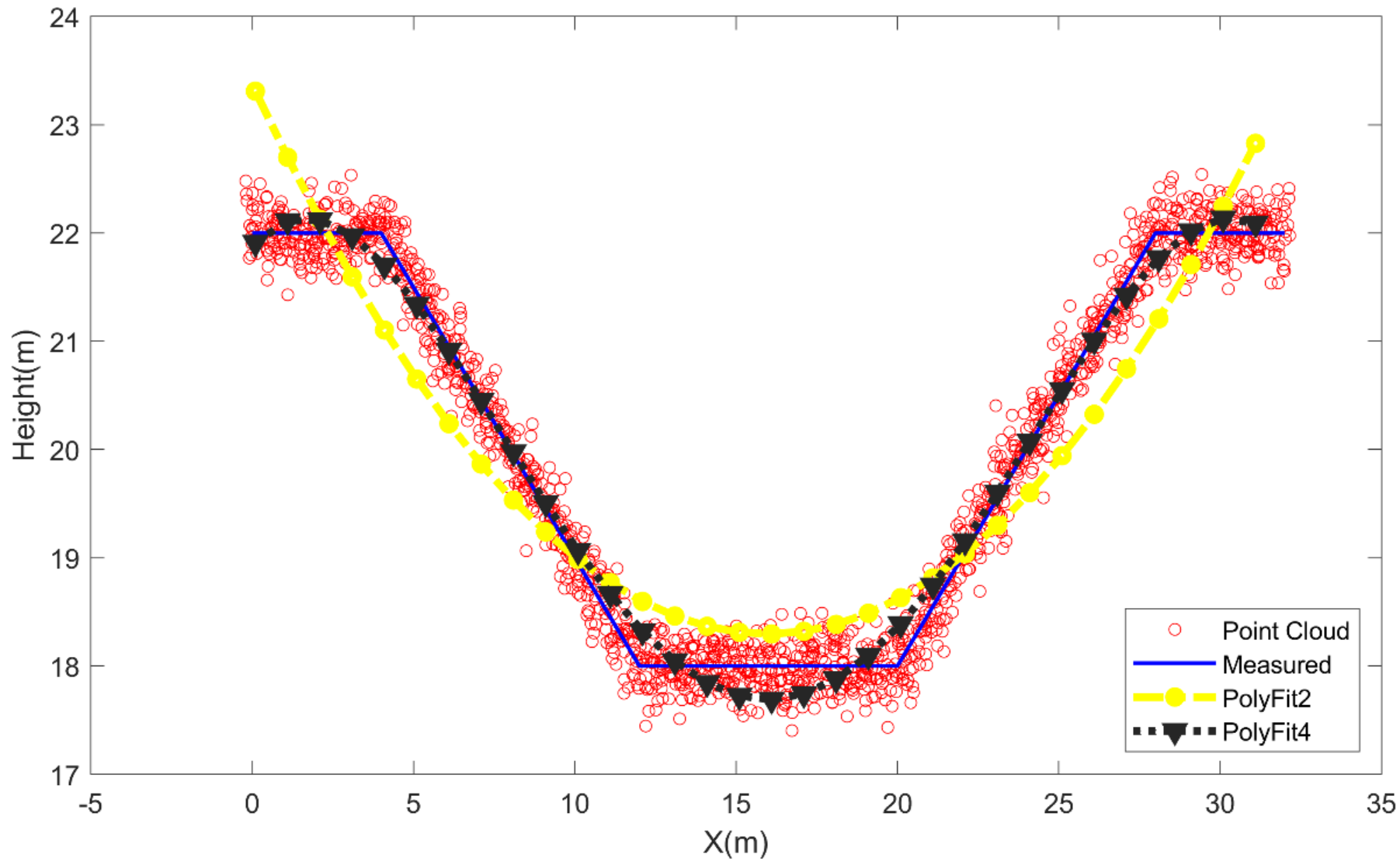

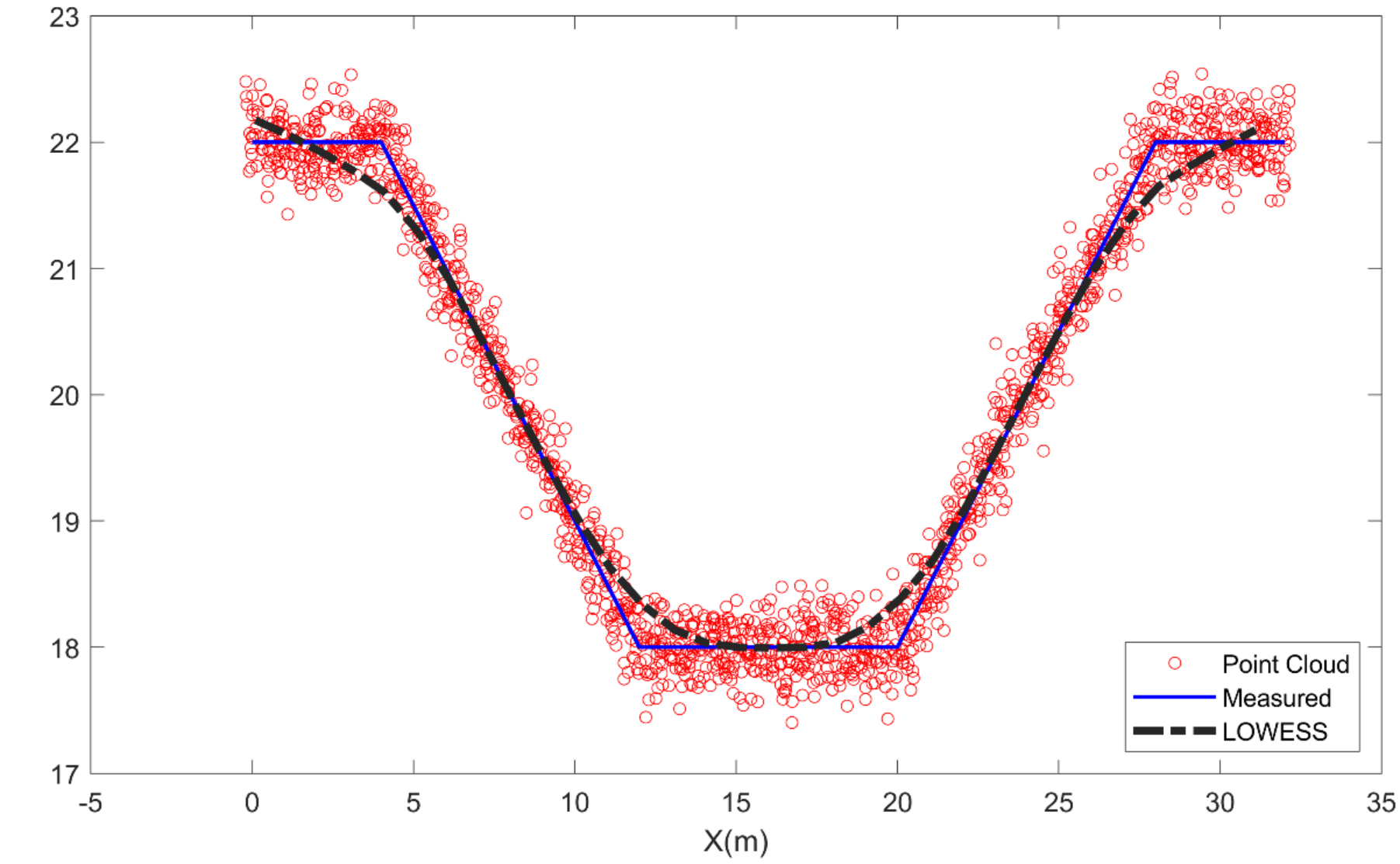

569
Figure 6. Polynomial regression (top panel) and LOWESS (bottom panel) were fitted to the simulated point cloud data of 10 times the assumed cloud point (red circles) and assumed trapezoidal channel (blue line).

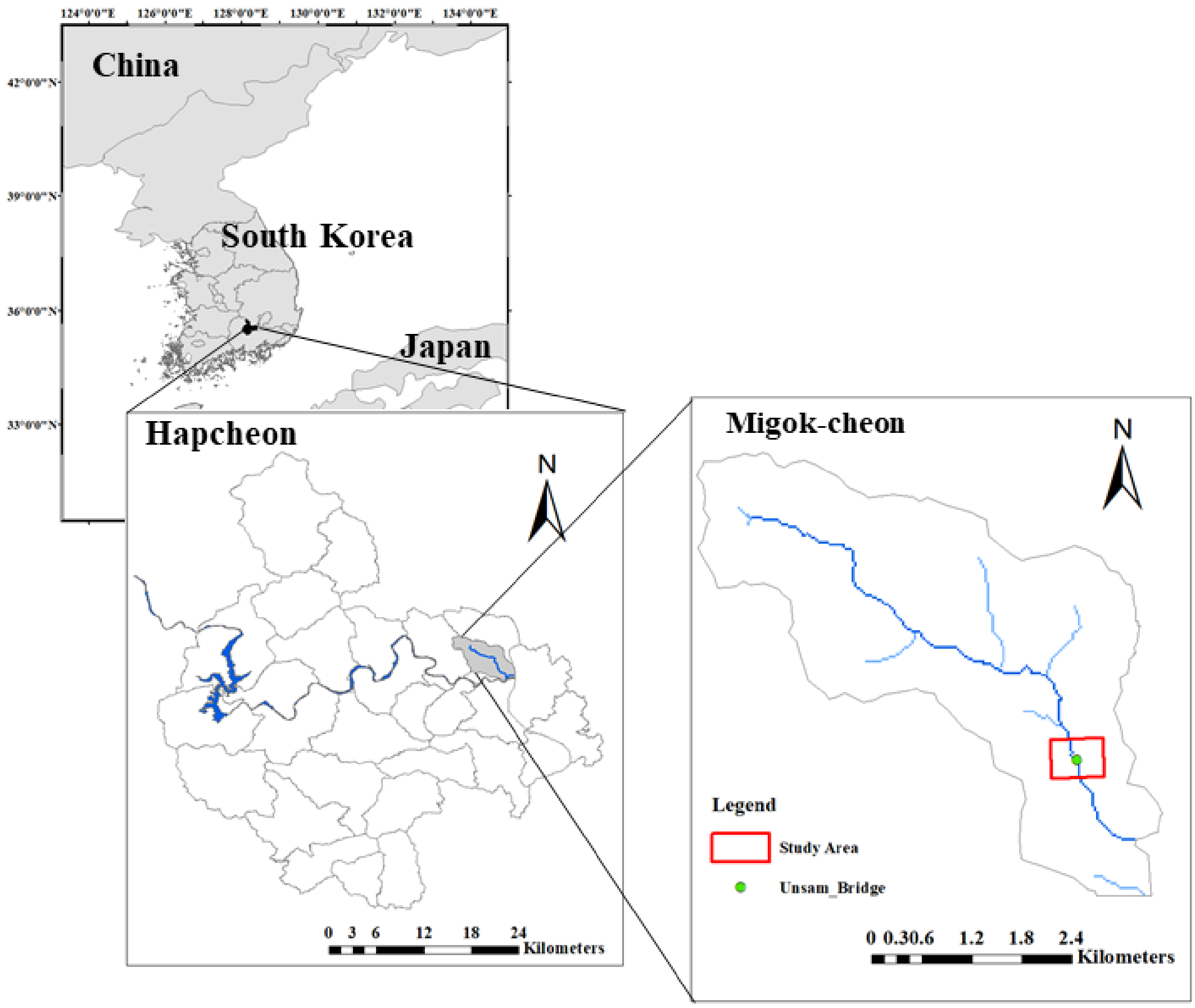

Figure 7. Study area of the applied stream, Migok-choen in South Korea, located in the province of Hapchoen-gun.

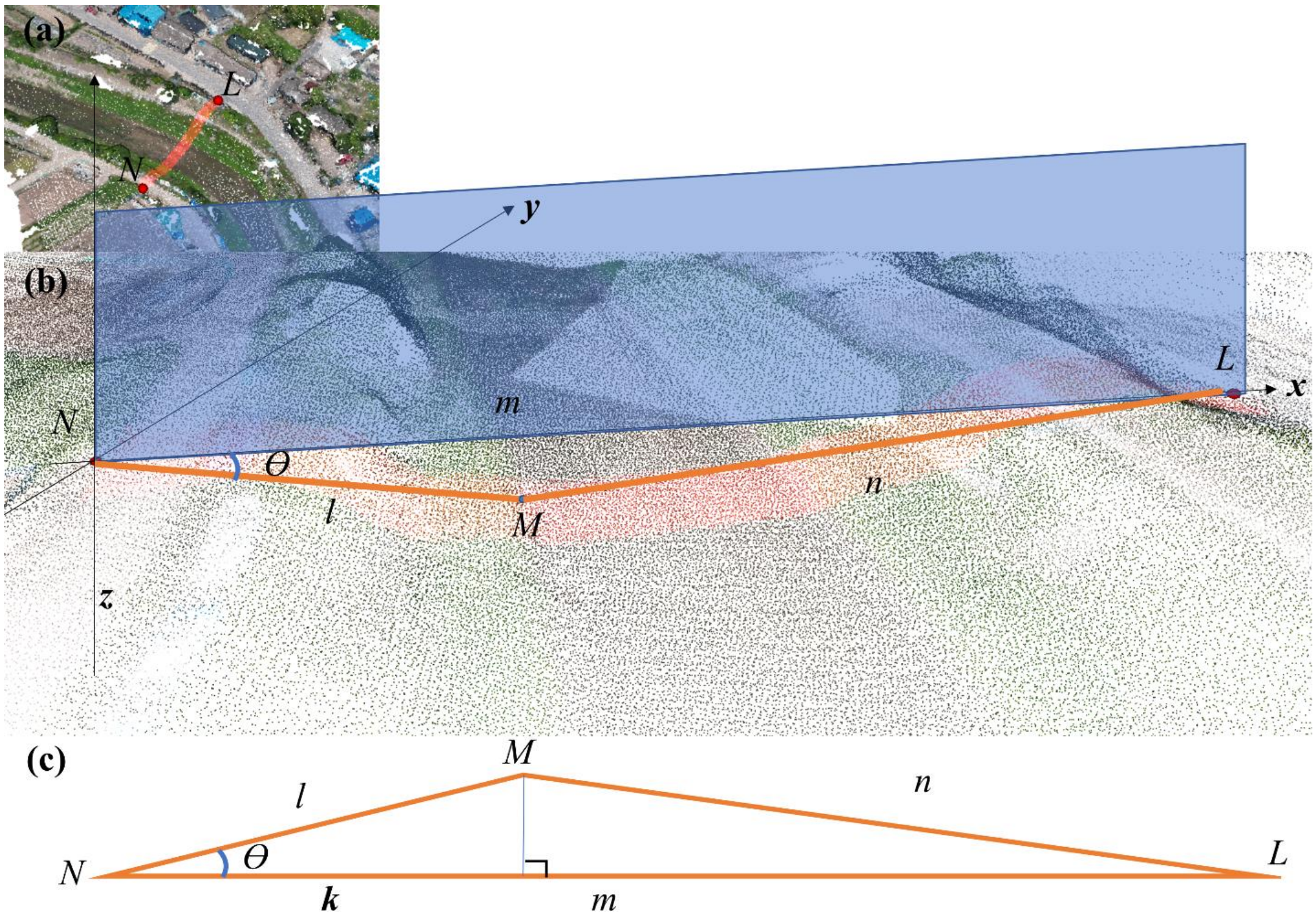

Figure 8. Example of the distance measurement: (a) aerial photo with a selected cross-section (two red dots, L and N, and thick red line); (b) magnified photo of Panel (a) with assisted 3D axis (x, y, and z) and the selected point (M); (c) emphasized triangle with the points of NML. Note that (1) the cross-section can be defined with the x-axis by connecting points N and L with the line; (2) the point M is the example point that contains the red line at Panel (a), which is a group of points in reality; and (3) the actual distance of M from N in the x-axis is represented as *k*, which can be designated as N to the point that meets line NL perpendicularly from M. The aerial images were taken from the authors.

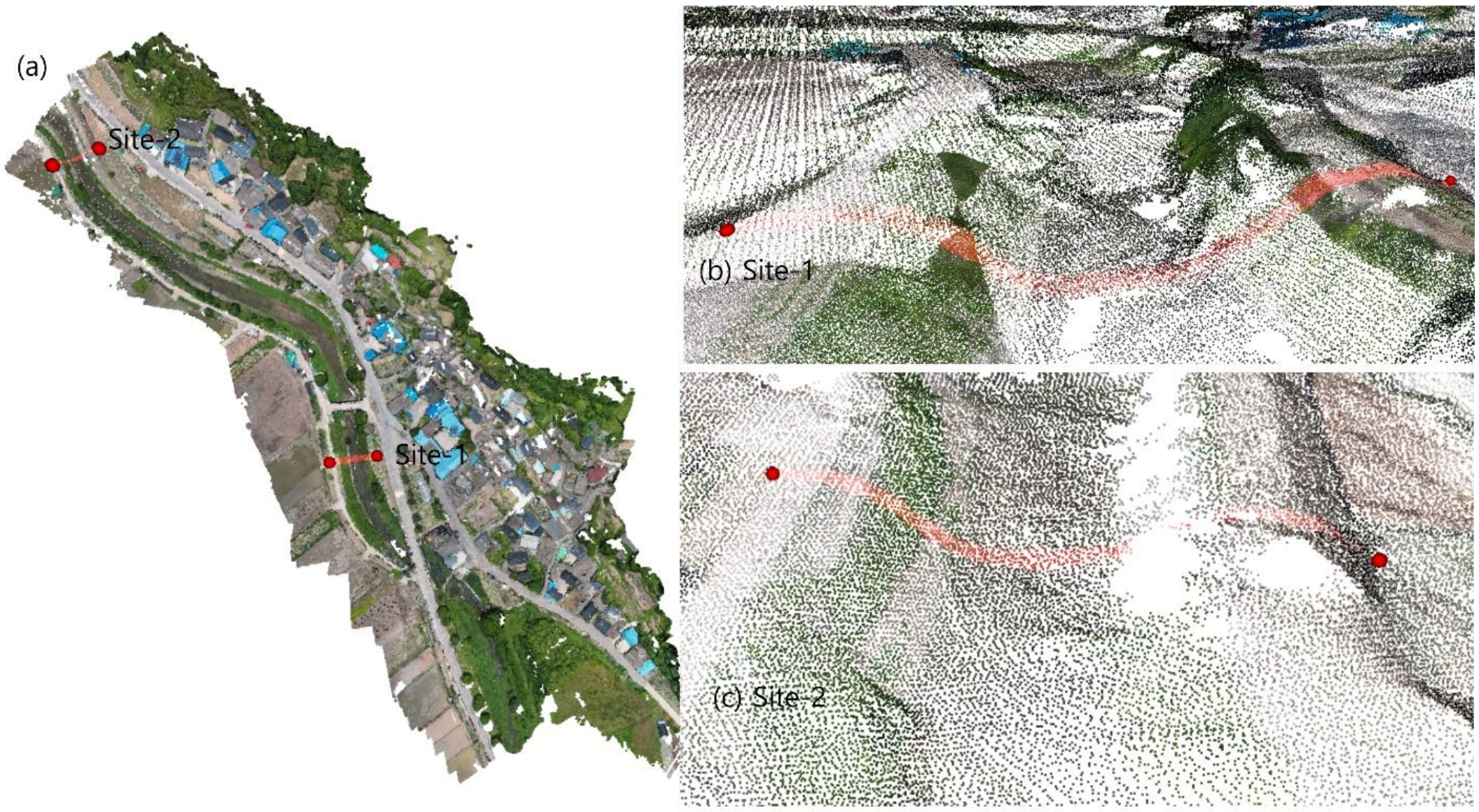

Figure 9. Two tested Sites in the Migok-cheon stream. Note that the right panels magnify the tested sites by showing the point clouds of the observed data taken from the UAV photographs. The aerial images were taken from the authors.

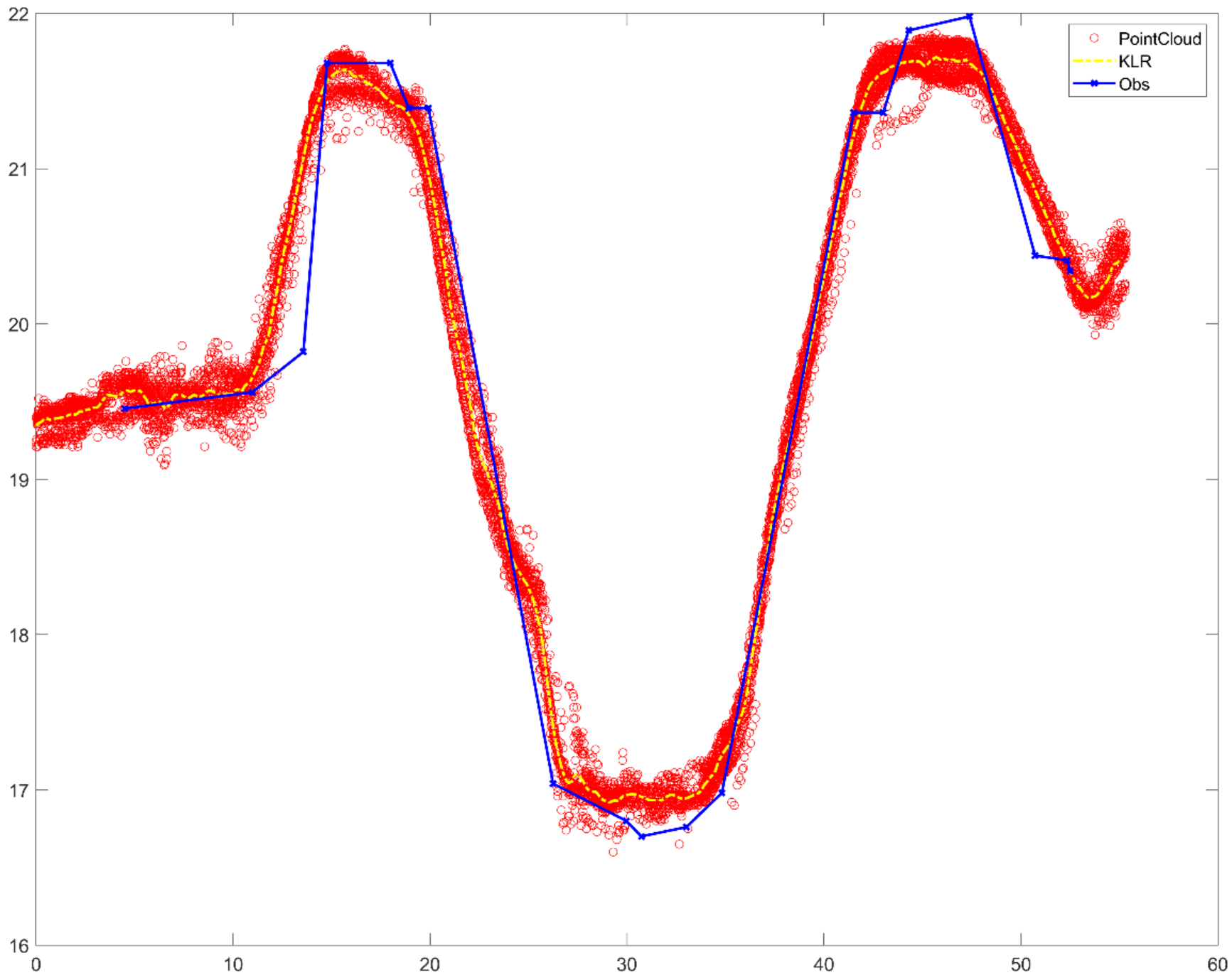

Figure 10. Point cloud data (red circles) and fitted KLR line (yellow dashed line) as well as the observed surveying for Site-1. Note that the observed line was drawn from the previous surveying in 2005.

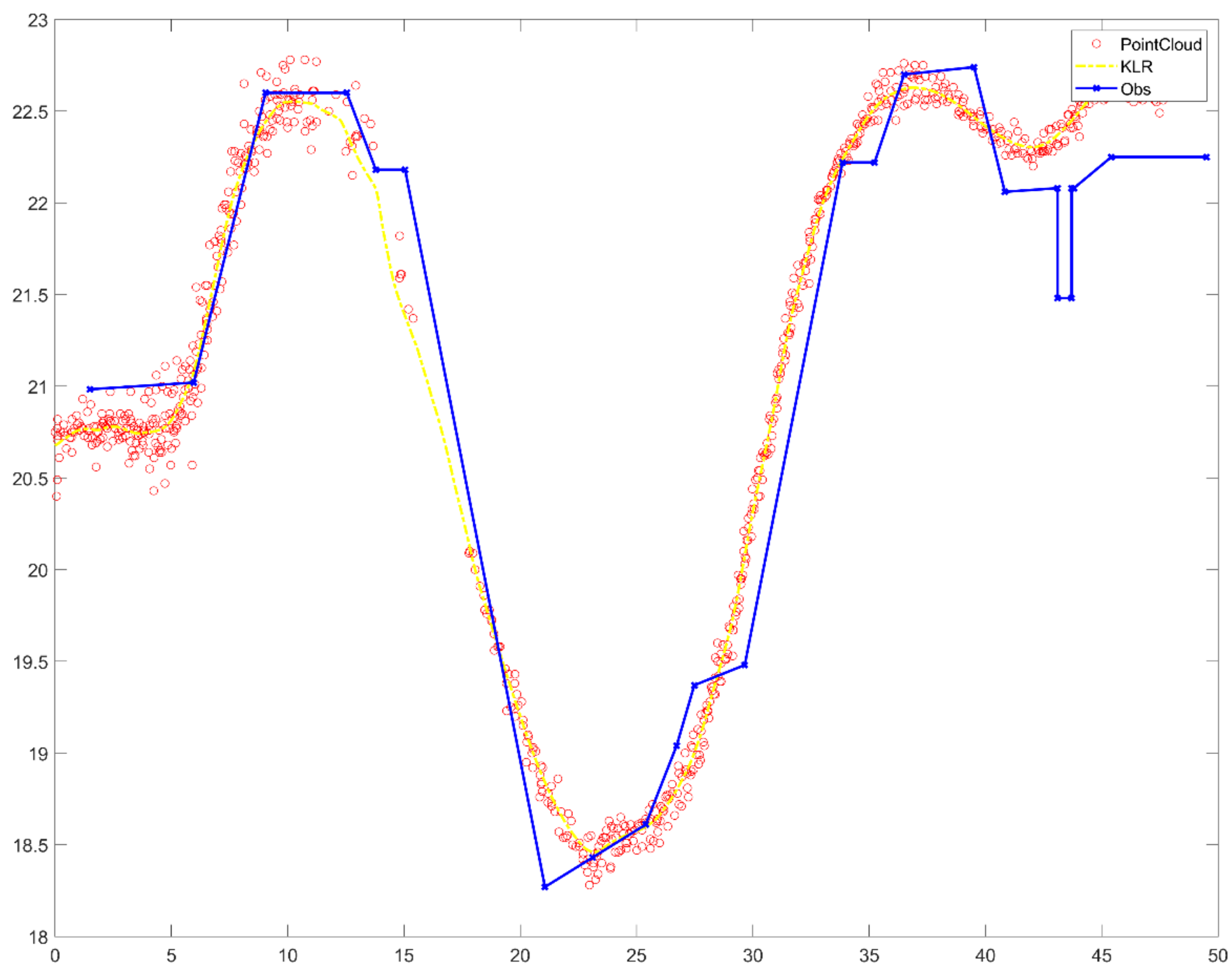

Figure 11. Point cloud data (red circles) and fitted KLR line (yellow dashed line) as well as the observed surveying for Site-2. Note that the observed line was drawn from the previous surveying in 2005.

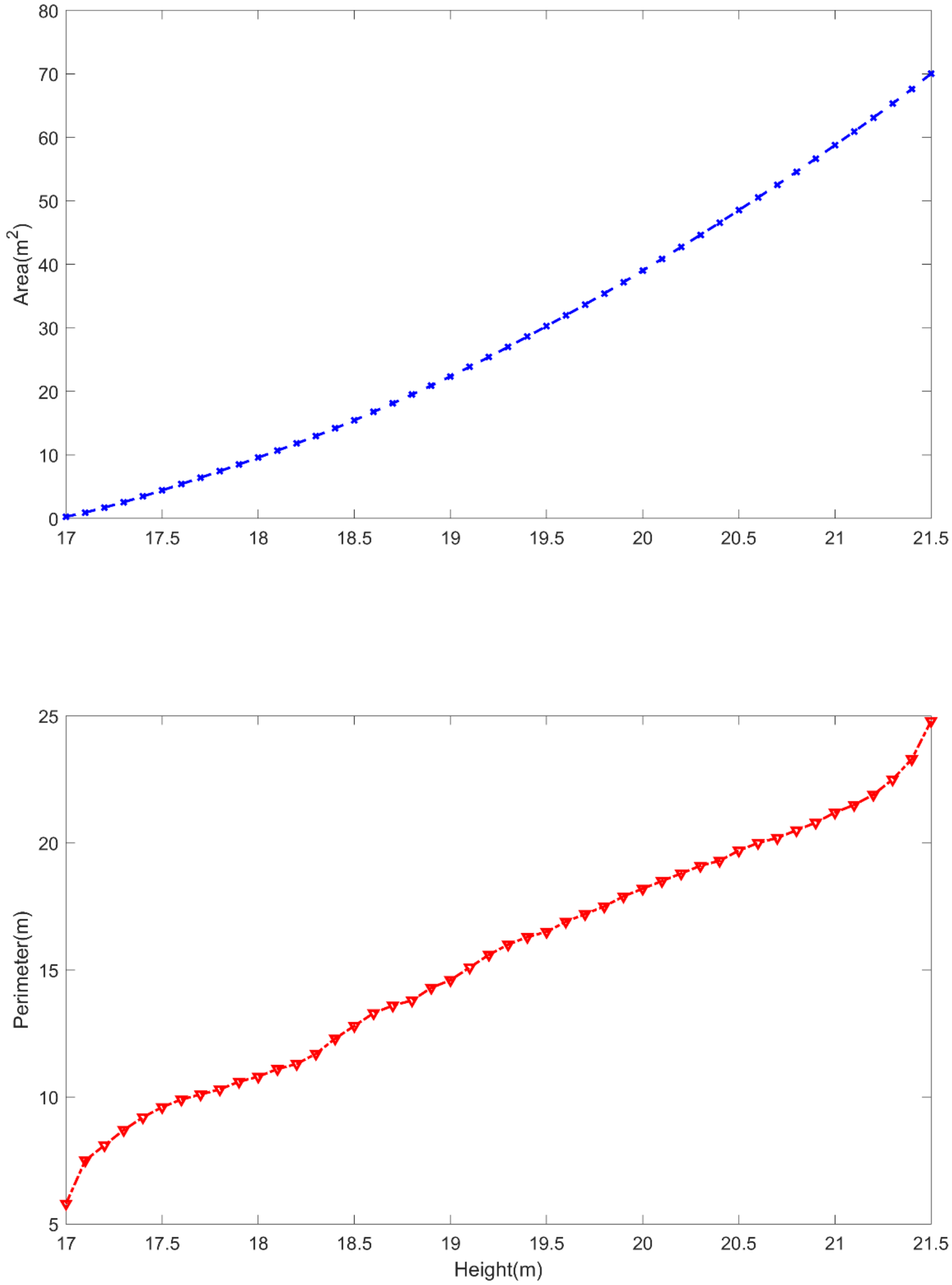

Figure 12. Estimated area and perimeter with the fitted KLR line (see the dashed yellow line of Figure 10) for Site-1 shown in Figure 7.

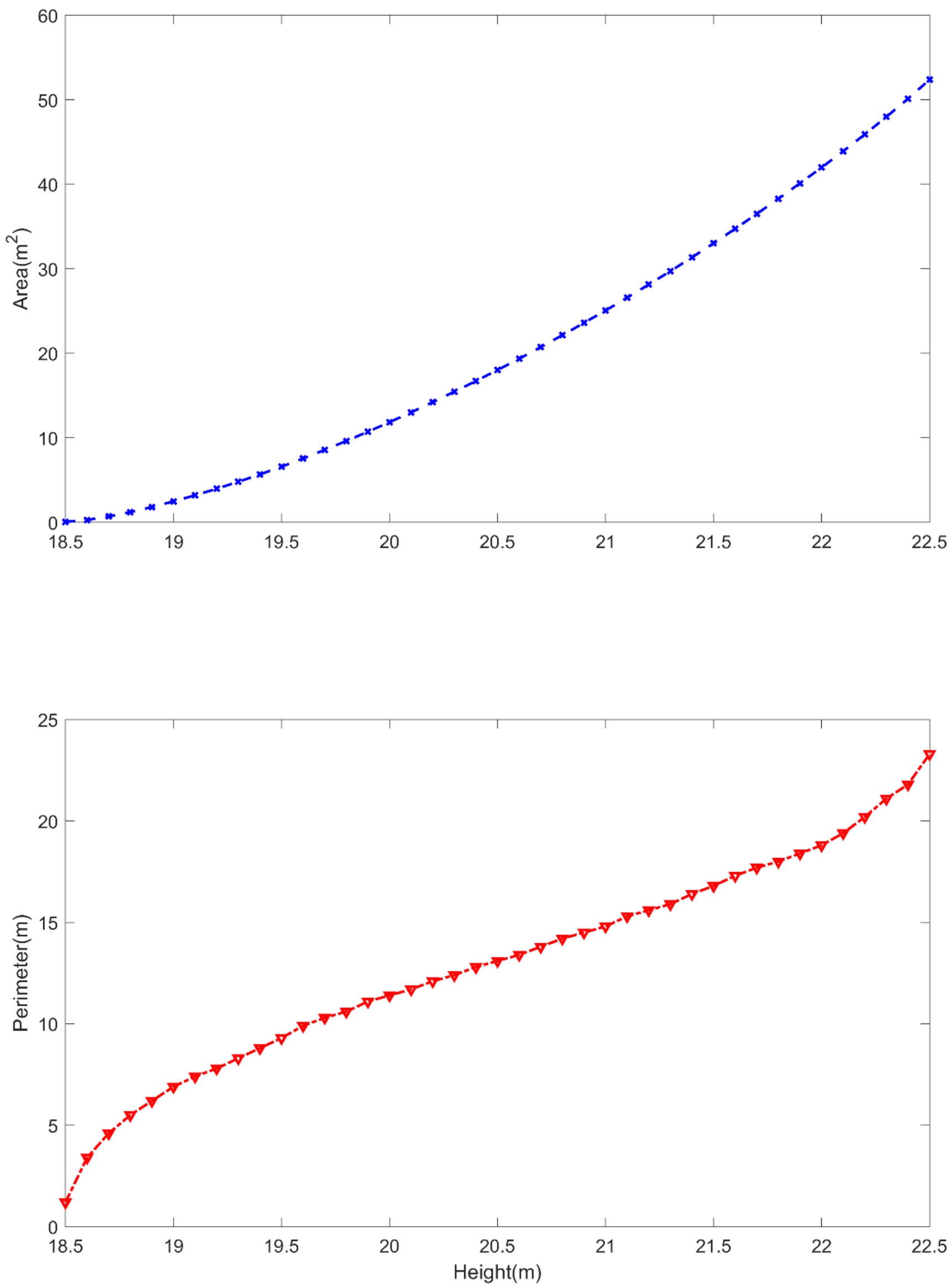

Figure 13. Estimated area and perimeter with the fitted KLR line (see the dashed yellow line of Figure 10) for Site-2 shown in Figure 7.