# Peer review of "Nonparametric-based estimation method for river cross-sections"

_Geoscientific Model Development, 2021_

## Author Comment (AC1)

Author response to the reviews of the paper "Nonparametric-based estimation method for river cross-sections with point cloud data from UAV photography URiver-X version 1.0 -methodology development

,,

(Manuscript # gmd-2021-309)

Overall:

The goal is worthy, but the methodology, reproducibility, validation, and results are questionable. I recommend rejecting the paper.

The authors appreciate this reviewer's comment. The authors tried their best to improve the quality of the manuscript following this reviewer's comment.

- 1) To generalize the proposed method, two typical shapes as V-shape and U-shape for a river channel were added from the other reviewer's comment and all three models as LOWESS, KLR, PolyFit models were applied to those models.
- 2) Two additional sites were added in the study area from this reviewer's comment and all four sites were applied to three tested models of LOWESS, KLR, PolyFit including the performance measurement of RMSE and MAE.
- 3) The authors agree that the general UAV sensor cannot penetrate the water surface and its application is limited. However, there are a number of streams that has no flow or close to zero during a dry season so that their cross-sections can be surveyed with a UAV photogrammetry. Even though no bathymetry data were tested in the current study, the authors consider that there are no reasons that this method cannot be applicable to the bathymetry data. The authors believe that this methodological development presented in the current manuscript might be beneficial and can be further generalized when the current proposed model has applied in the field.
- 4) The authors invite one professional in the hydraulic field as an author, Prof. Vijay P. Singh, to further revise and edit the manuscript. Prof. Singh made a significant contribution to revise and edit the manuscript. Hope this inclusion of an author is acceptable.
- No discussion of UAV type used (sensor, height flown, date flown, resolution), which is essential to place this work within the vase literature devoted to UAV-based survey.

Reply: The authors appreciate this reviewer's detailed comment and totally agreed with this comment. Before, the authors had considered that this information might not be necessary for this application because the proposed model can be easily applied and not restricted to those elements. According to this reviewer's comment, the discussion on the UAV type was added in the manuscript. Hope this modification is satisfactory to this reviewer. "Aerial photos over the selected Migok-cheon were obtained with the unmanned aerial vehicle (also termed drone), DJI Phantom 4. This UAV is one of the most popular professional drones on the market and contains an advanced stereo vision positioning system that provides precise hovering even without satellite positioning support (Hamdi et al., 2019). The camera applied is FC3411 with ISO-110 and the image sensor of 1/2.3" CMOS, and the images taken from DJI Phantom 4 are 5472x3648 pixels at approximately 10 M with the horizontal and vertical resolutions of 75 DPI. Pix4Dcapture was employed to map the target area. The flight with a height of 75 meters was made on July 08, 2021."

• UAV cannot penetrate water depth, unless using bathymmetric technology (which was not discussed here). Therefore, the significant matching of the point cloud data with the engineered surveying data in the Figures is questionable.

Reply: The authors appreciate this reviewer's insightful comment. The authors agree with this comment. UAV cannot penetrate water depth without additional technique. The authors are thankful for guiding what the authors missed before getting into detail. Accordingly, the authors point out the limitation of the proposed method and discuss how the water portion should be interpreted. For the engineered surveying data, the author considers that the tested stream is a dry stream and water depth is close to zero except the wet season (June-October). Hope this explanation acceptable to this reviewer.

• Why is the focus of this paper on channel cross-sections when no hydrological conditions are described, modeled, or assessed in terms of stream flow? In essence, the approach mentioned here can be used for any form of landscape survey analysis, and in fact, since the lack of UAV's ability to penetrate water surface is a limitation that was not adequately addressed in this study, the overall paper study should probably focus on general landscape patterns or dry streams for comparison.

Reply: The authors are thankful to this reviewer's comment. The authors have worked at the hydrological field and modeled streams in South Korea. Most of streams in South Korea are dry except the wet season (June-October) since their watersheds are small and during the dry seasons, streams are without much flow. Also, UAV surveying has been studied in South Korea to surrogate ground surveying and provide additional information. Therefore, the authors consider that this application of UAV photogrammetry of river cross-sections can be beneficial to engineers and researchers even though there are limitations on UAV ability to penetrate water surface. Also, generalization of the proposed method might request extensive studies with a number of different types of surfaces and therefore, the authors include two more synthetic simulation data as U-shape and V-shape as well as two more sites for the case study. In addition, the LOWESS and PolyFit models were applied to all the case studies. Hope this discussion and additional work can be acceptable to this reviewer.

Specific Comments:

Lines 62-63 - This is based on UAV of wet versus dry pixels and not stream cross-section. Flood inundation from UAV is a well-established field. Stream cross-sections are not commonly used with UAV, as the sensors cannot penetrate the water depth and gather data of the full wetted perimeter. Traditional surveying is essential to supplement UAV or LiDAR-based point clouds for purposes of hydraulic modeling.

Reply: The authors appreciate this reviewer's pinpointing comment and agree with this comment in that traditional surveying is essential to supplement UAV point clouds. Following this reviewer's comment, the sentence was removed. In addition, the authors consider that UAV surveying have a considerable potential to apply it to river cross-sections since not only penetrating water surface technique is developing but there are a number of sites that are normally dry in dry season especially in South Korea with smaller watersheds. This has been discussed in the discussion section of the current manuscript accordingly as well as the conclusion.

"The results of the synthetic simulation study and the case study present that the proposed KLR model can demarcate the cross-sections of a river with different shapes. However, there are some limitations and conditions to apply the proposed model in the demarcation of river cross-sections. At first, UAV sensors cannot penetrate water depth unless bathymetric technology is not applied. Currently, river photogrammetry with bathymetry data has been applied to penetrate water body using specialized sensors, such as Light Detection and Ranging (LiDAR), which is called bathymetry LiDAR (Allouis et al., 2010; Fernandez-Diaz et al., 2014). The case study of the current study does not use the bathymetry data, since the water depth is very shallow and not critical to illustrate a river cross-section. The proposed KLR model with the point cloud data must be carefu lly applied to a dry stream or very shallow river with the water surface whose level is ignorable e specially for its discharge amount. Otherwise, a bathymetry data must be applied using a special sensor (e.g. bathymetry LiDAR). "

Lines 81-85 - A DEM, similarly, cannot penetrate water depth. In most DEM approaches, the depth of the stream still must be "burned" into the digital model, which requires knowing the depth a priori.

*Reply: The authors appreciate this reviewer's insightful comment. We discussed the limitation of the proposed technique especially not penetrating water depth as mentioned in the discussion section. Hope this modification is satisfactory to this reviewer.*

Lines 87-88 - Where do you demonstrate the strengths of higher-resolution points, compared to DEM resolutions from satellites, for purposes of hydraulic modeling? Generally, a very rough depiction of channel top width, bottom width, side slope, and Manning's n roughness is adequate for catchment-scale modeling in software such as HEC-RAS, since the water conservation equations are not highly sensitive to small changes in channel geometry.

Reply: The authors appreciate this reviewer's critical comment and totally agree with this comment that a rough depiction of river cross-section might be adequate for catchment-scale modeling. The authors do not compare the cloud point-driven data to DEM from satellite. However, the authors consider that the ground surveying also has its own limitation such as the ground surveying points are limited only to accessible locations by foot and time and cost

consuming procedure. Therefore, the authors consider that UAV surveying can also be a potential surrogate for its relatively cheap and time-saving. Furthermore, the river cross-section with UAV surveying can also be beneficial when the ground surveying has not been made and further resources cannot be available for additional ground surveying. In this case, the river cross-section with UAV surveying can extract any places inside the surveyed area. Also, detailed description of river cross-section might be helpful in other applications since related UAV techniques have been developed fast. This has been discussed in the discussion section. Hope this modification is satisfactory to this reviewer.

Lines 90-93 - Natural river cross-sections do not typically form a trapezoidal shape, but rather a U-shape, due to erosion and sediment redeposition throughout the channel. Manmade channels are typically trapezoidal. It is unclear throughout the study if the sample stream is manmade or natural. I believe it is manmade, due to having engineered plans and appearing trapezoidal and straightened in the aerial figures, but little background information on this stream is able to be found online. Moreover, having UAV produce reliable riverine cross-sections in ungauged and natural areas is most useful for modeling hydrology in remote systems, as engineered systems have plans and field surveys available. This paper claims to address this gap but then, it appears, uses a man-made channel for validation.

Reply: The authors are thankful to this reviewer's comment for pointing out the authors' mistake. The applied river section is natural river cross-sections. Note that about 80% of national rivers and over 40% of local rivers over South Korea are manmade by revetment and the shape of manmade rivers are in trapezoidal. The application must be denoted to manmade river as mentioned by this reviewer's comment. The manuscript was modified accordingly as follows. Furthermore, A natural type of U-shape cross-section as well as V-shape was synthetically generated and applied in the simulation study. Hope this modification and additional work are satisfactory to this reviewer.

"Therefore, the current study proposes a demarcation technique for river cross-sections from the point clouds of UAV aerial surveying especially in a small study area. For example, about 80% of national rivers and over 40% of local rivers are maintained by the construction of dikes and revetments for flood control in South Korea. The shape of manmade rivers is mostly trapezoidal due to the stability and easy discharge. A cross-section of manmade rivers also often contains abrupt changes and small bumps as well as smooth variations from aging in natural rivers. The demarcation technique must reproduce the characteristics of manmade channels as well as the ones of typical rivers from aging in natural channels. The proposed demarcation model based on the KLR model was tested to determine whether to reproduce those characteristics."

"3.3.1 U-shape cross-section

 $\overline{z}$

*U-shape cross-section that are close to a natural river was tested. The U-shape cross was modelled with a power function from Neal et al. (2015) as*

$$w_f = w_F \left(\frac{h_f}{h_F}\right)^{\overline{s}}$$
(13)
$$h_f = h_F \left(\frac{w_f}{w_F}\right)^{\overline{s}}$$
(14)

where  $w_f$  indicate the flow width while  $w_F$  is the bank-full flow width and  $h_f$  and  $h_F$  is the height of flow width and the height in a bank-full condition, respectively. Also, s is the parameter to vary

the shape of the cross-section. Here, s=5 was set as used in Neal et al. (2015) as the basic value. To design a similar bank in the trapezoid model in the previous test,  $h_F=5m$  and  $w_F=20m$  was used. The number of points for the U-shape cross-section is 262 points including the flat river bank and the designed cross-section was presented in a blue solid line with cross markers as shown in Figure 7. The synthetic point cloud data was simulated with Eq.(11) and the number of point clouds are 10 times of the U-shape cross-sections (i.e. 2620 points), shown with the red dots in Figure 7.

This designed U-shape cross-section was fitted to the proposed KLR model and the other models as LOWESS and PolyFit and shown in Figure 7. Note that a=2 (see Eq.(10)) was applied for the KLR model from the result of the trapezoid case. The result in Figure 7 indicates that the KLR model matched well the U-shape cross-section without any deviation. Meanwhile, the LOWESS model fitted U-shape cross-section well in the middle part, but the connected part of the U-shape cross-section was not fitted well. The PolyFit model fairly fitted the U-shape cross-section with the  $4^{th}$  model (i.e. PolyFit4) except slight deviation in the connected area between the slope and top. The PolyFit2 and PolyFit3 are poorly performed due to its limit of the flexibility."

3.3.2 V-shape cross-section

One of the unique shape of cross-sections is the V-shape for a river cross-section. The V-shape weir (or triangle shape, v-notch) was often built to provide a highly accurate solution for open channel flow measurement. Also, the V-shape river cross-section can be developed naturally when the sides are cut down and attacked by weathering. In addition, the loosened material slowly creeps down the slope by gravity. A V-shape cross-section was synthetically designed as shown in Figure 8 with the height of 4m and the top width of 16m so that the slopes of both sides are in 1:2. The cross-section was divided to 121points, including the flat river bank shown with the blue solid line with cross markers in Figure 8 and 10 times of the points was synthetically generated for point cloud data with Eq. (11) and presented with the red dots.

The point cloud data was fitted to KLR, LOWESS, and PolyFit models and shown in the panels (a), (b) and (c) of Figure 8, respectively. Here, a=2.0 was also employed for the KLR model. The result of the KLR model indicates that the V-shape cross-section also was fitted well by the KLR model with a minimal deviation at the acute angle bottom section. Meanwhile, the LOWESS model highly deviated at the acute bottom section and slight deviation was present at the top connected part. The PolyFit model did not fairly fit the V-shape model even with PolyFit4. Further higher order model was tested (i.e. PolyFit5 and PolyFit6) and no improvement was found with increasing the order for the PolyFit model.

Table 1 presents the estimated RMSE and MAE for three tested models of KLR, LOWESS, and PolyFit4 with trapezoidal, U-shape, and V-shape cross-section data. Note that only PolyFit4 was presented, since 4th-degree was the best for the PolyFit models. The RMSE and MAE estimates present that the KLR model outperforms the other fitting models, while the other two models of PolyFit4 and LOWESS are comparable to each other for trapezoidal and U-shape cross-sections. For V-shape channel, the LOWESS much better performed than the PolyFit4, since the PolyFit4 is a parametric model that connects the points rather smoothly and abrupt change cannot be modelled well due to its limited flexibility. Overall, the simulation study indicates that the proposed KLR model is a good alternative to demarcate the different shape cross-sections.

Further, nonparametric models and other regression models, such as logistic regression (Ahmad et al., 1988; Elek and Márkus, 2004; Orlowsky et al., 2010; Simonoff, 1996), can be tested. However, the simulation study with the trapezoid channel that is similar to the real river cross-section shows that the presented KLR nonparametric model originally developed by Lee et al.

(2017) is suitable for demarcating the cross-section of a river. The major reason for the good performance is that the KLR model employs only k-nearest neighbor observations. This approach might not be beneficial, when an overall trend is needed and not enough observations are available. However, the point cloud data taken from UAV aerial surveying often provides a large enough number of points in the data set. Furthermore, the cross-sections in a manmade river can contain irregularity and abrupt changes by river aging. This feature can be captured only through fitting nearby observations. Therefore, the KLR model might be a suitable alternative to demarcating the cross-section of a river with the cloud point dataset."

Figure 7. Synthetic U-shape river cross-section (blue solid line with cross markers) and the simulated point could data (red circles) of 10 times the synthetic channel (2620 points total) with Eq.(17) as well as the fitted estimates to KLR (the panel(a)), LOWESS (the panel(b)), and PolyFit (the panel(c)). Note that the U-shape river cross-section was designed with the power function as in Eqs. (19) and (20) and the U-shape was synthetically built following the reference of Neal et al. (2015) and the section was divided into 262 points.

---

## Author Comment (AC2)

Author response to the reviews of the paper
"Nonparametric-based estimation method for river cross-sections with point cloud data from
UAV photography URiver-X version 1.0 -methodology development
"

(Manuscript # gmd-2021-309)

This is a review of the manuscript "Nonparametric-based estimation method for river cross-sections with point cloud data from UAV photography URiver-X version 1.0 -methodology development" by Lee and Sung. The study explores the suitability of K-nearest neighbor local linear regression for estimating cross-sectional geometry from digital images collected by UAV. The K-nearest neighbor has been used extensively for interpolating/extracting information from point datasets. Also, the study has only implemented the methodology on two cross-sections at the same reach with similar river geometry characteristics making any conclusion highly localized and not generalizable in other locations.

I recommend rejecting the manuscript. What the authors need to do find multiple areas with different geomorphological characteristics and available bathymetry data, survey them using UAV and analyze the different regression techniques at multiple locations to come up with generalizable conclusions.

*Reply: The authors appreciate this reviewer's comment. The authors tried their best to improve the quality of the manuscript following this reviewer's comment.*

1) *The KNN model has been widely employed in point datasets. However, the KLR model adopted in the current study was developed by the first author in Lee et al. (2017). This has not been much studied in point cloud dataset. The authors clearly mentioned how the model was developed and has been applied to different datasets so far in the current version of the manuscript.*

2) *To generalize the proposed method, two typical shapes as V-shape and U-shape for a river channel were added and all three models as LOWESS, KLR, PolyFit models were applied to those models.*

3) *Two additional sites were added in the study area and all four sites were applied to three tested models of LOWESS, KLR, PolyFit including the performance measurement of RMSE and MAE.*

4) *The authors agree that extensive locations with different geomorphological characteristics including bathymetry data should be applied to generalize the proposed model. However, there are a number of streams that has no flow or close to zero during a dry season so that their cross-sections can be surveyed with a UAV photogrammetry. Even though no bathymetry data were adopted, the authors consider that there are no reasons that this method cannot be applicable to the bathymetry data. In order to cover different shapes and more natural river shapes, the authors include V-shape and U-shape synthetic cross-sections. Please note that collecting the bathymetry data and different geomorphological stations cost significant amount of fund and time. The authors believe*

*that this methodological development presented in the current manuscript might be beneficial and can be further generalized when the current proposed model has applied in the field. The authors consider that it does not seem capable prove everything with collecting all possible shapes in the current study. Hope this reviewer understand this circumstance and satisfactory to this modification.*

5) *The authors invite one professional in the hydraulic field as an author, Prof. Vijay P. Singh, to further revise and edit the manuscript. Prof. Singh made a significant contribution to revise and edit the manuscript. Hope this inclusion of an author is acceptable.*

Major comments:

- UAV can only capture the water surface and not the actual bed if the river reach is not dry. Therefore, UAV surveys are more suited for ephemeral/dry river reaches or for capturing the overbank/floodplain geometry. For perennial rivers, additional information is required (either velocity measurements or bathymetry measurements) to further refine the cross-sections. The authors should clarify in Introduction what the intended application is for such cross-sections in this study and also note the drawbacks of UAV.

   *Reply: The authors appreciate this reviewer's pinpointing comment. The authors discussed clearly the limits and conditions of the current study in the discussion section. The authors consider that the discussion section might be better to deal with the limits and conditions of the propose method. Also, the introduction was modified by explaining the focus of the current study, accordingly.*

- Most of the study hinges on the premise that river cross-sections are trapezoidal which is not completely true. The assumption proposed by Chow (1959) is only reasonable in areas with sparse data and not where detailed surveys have been carried out. River channels exhibit a wide variety of shapes, and depending on the intended application, they may need to be captured in lot of detail. As such, the authors should do the analysis for idealized cross-sections with different shapes such as triangular, trapezoidal, and parabolic shapes to see how the different regression techniques compare.

   *Reply: The authors appreciate this reviewer's insightful comment and totally agree with this reviewer's comment. The authors made some modifications throughout the manuscript to refer this comment. Following this comment, the authors include two additional synthetic shapes in the simulation study as U-shape for more natural rivers and V-shape. The U-shape function was built with the power function suggested by Neal et al. (2015). Hope that this additional study is satisfactory to this reviewer.*

*"The results of the synthetic simulation study and the case study present that the proposed KLR model can demarcate the cross-sections of a river with different shapes. However, there are some limitations and conditions to apply the proposed model in the demarcation of river cross-sections. At first, UAV sensors cannot penetrate water depth unless bathymetric technology is not applied. Currently, river photogrammetry with bathymetry data has been applied to penetrate water body using specialized sensors, such as Light Detection and Ranging (LiDAR), which is called bathymetry LiDAR (Allouis et al., 2010; Fernandez-Diaz et al., 2014). The case study of the current*

study does not use the bathymetry data, since the water depth is very shallow and not critical to illustrate a river cross-section. The proposed KLR model with the point cloud data must be carefully applied to a dry stream or very shallow river with the water surface whose level is ignorable especially for its discharge amount. Otherwise, a bathymetry data must be applied using a special sensor (e.g. bathymetry LiDAR)."

"3.3.1 U-shape cross-section
U-shape cross-section that are close to a natural river was tested. The U-shape cross was modelled with a power function from Neal et al. (2015) as

$$w_f = w_F \left(\frac{h_f}{h_F}\right)^{\frac{1}{s}} \qquad (13)$$

$$h_f = h_F \left(\frac{w_f}{w_F}\right)^{s} \qquad (14)$$

where $w_f$ indicate the flow width while $w_F$ is the bank-full flow width and $h_f$ and $h_F$ is the height of flow width and the height in a bank-full condition, respectively. Also, s is the parameter to vary the shape of the cross-section. Here, s=5 was set as used in Neal et al. (2015) as the basic value. To design a similar bank in the trapezoid model in the previous test, $h_F$=5m and $w_F$=20m was used. The number of points for the U-shape cross-section is 262 points including the flat river bank and the designed cross-section was presented in a blue solid line with cross markers as shown in Figure 7. The synthetic point cloud data was simulated with Eq.(11) and the number of point clouds are 10 times of the U-shape cross-sections (i.e. 2620 points), shown with the red dots in Figure 7.
This designed U-shape cross-section was fitted to the proposed KLR model and the other models as LOWESS and PolyFit and shown in Figure 7. Note that a=2 (see Eq.(10)) was applied for the KLR model from the result of the trapezoid case. The result in Figure 7 indicates that the KLR model matched well the U-shape cross-section without any deviation. Meanwhile, the LOWESS model fitted U-shape cross-section well in the middle part, but the connected part of the U-shape cross-section was not fitted well. The PolyFit model fairly fitted the U-shape cross-section with the 4[th] model (i.e. PolyFit4) except slight deviation in the connected area between the slope and top. The PolyFit2 and PolyFit3 are poorly performed due to its limit of the flexibility."
3.3.2 V-shape cross-section
One of the unique shape of cross-sections is the V-shape for a river cross-section. The V-shape weir (or triangle shape, v-notch) was often built to provide a highly accurate solution for open channel flow measurement. Also, the V-shape river cross-section can be developed naturally when the sides are cut down and attacked by weathering. In addition, the loosened material slowly creeps down the slope by gravity. A V-shape cross-section was synthetically designed as shown in Figure 8 with the height of 4m and the top width of 16m so that the slopes of both sides are in 1:2. The cross-section was divided to 121points, including the flat river bank shown with the blue solid line with cross markers in Figure 8 and 10 times of the points was synthetically generated for point cloud data with Eq. (11) and presented with the red dots.
The point cloud data was fitted to KLR, LOWESS, and PolyFit models and shown in the panels (a), (b) and (c) of Figure 8, respectively. Here, a=2.0 was also employed for the KLR model. The result of the KLR model indicates that the V-shape cross-section also was fitted well by the KLR model with a minimal deviation at the acute angle bottom section. Meanwhile, the LOWESS model highly deviated at the acute bottom section and slight deviation was present at the top connected part. The PolyFit model did not fairly fit the V-shape model even with PolyFit4. Further higher

*order model was tested (i.e. PolyFit5 and PolyFit6) and no improvement was found with increasing the order for the PolyFit model.*

*Table 1 presents the estimated RMSE and MAE for three tested models of KLR, LOWESS, and PolyFit4 with trapezoidal, U-shape, and V-shape cross-section data. Note that only PolyFit4 was presented, since 4th-degree was the best for the PolyFit models. The RMSE and MAE estimates present that the KLR model outperforms the other fitting models, while the other two models of PolyFit4 and LOWESS are comparable to each other for trapezoidal and U-shape cross-sections. For V-shape channel, the LOWESS much better performed than the PolyFit4, since the PolyFit4 is a parametric model that connects the points rather smoothly and abrupt change cannot be modelled well due to its limited flexibility. Overall, the simulation study indicates that the proposed KLR model is a good alternative to demarcate the different shape cross-sections.*

*Further, nonparametric models and other regression models, such as logistic regression (Ahmad et al., 1988; Elek and Márkus, 2004; Orlowsky et al., 2010; Simonoff, 1996), can be tested. However, the simulation study with the trapezoid channel that is similar to the real river cross-section shows that the presented KLR nonparametric model originally developed by Lee et al. (2017) is suitable for demarcating the cross-section of a river. The major reason for the good performance is that the KLR model employs only k-nearest neighbor observations. This approach might not be beneficial, when an overall trend is needed and not enough observations are available. However, the point cloud data taken from UAV aerial surveying often provides a large enough number of points in the data set. Furthermore, the cross-sections in a manmade river can contain irregularity and abrupt changes by river aging. This feature can be captured only through fitting nearby observations. Therefore, the KLR model might be a suitable alternative to demarcating the cross-section of a river with the cloud point dataset."*

[Figure]

*Figure 7. Synthetic U-shape river cross-section (blue solid line with cross markers) and the simulated point could data (red circles) of 10 times the synthetic channel (2620 points total) with*

*Eq.(17) as well as the fitted estimates to KLR (the panel(a)), LOWESS (the panel(b)), and PolyFit (the panel(c)). Note that the U-shape river cross-section was designed with the power function as in Eqs. (19) and (20) and the U-shape was synthetically built following the reference of Neal et al. (2015) and the section was divided into 262 points.*

[Figure]

*Figure 8. Synthetic V-shape river cross-section (blue solid line with cross markers) and the simulated point could data (red circles) of 10 times the synthetic channel (2620 points total) with Eq.(17) as well as the fitted estimates to KLR (the panel(a)), LOWESS (the panel(b)), and PolyFit (the panel(c)). Note that (1) the V-shape river cross-section was designed with the height of 4 m and top width of 16 m and the section was divided into 121points.*

- The case-studies are highly specific cases in what looks like an engineered channel. It is imperative for the authors to incorporate more study sites with varied geomorphologic characteristics to make this study generalizable. At the moment, it looks like the conclusions are only true for that one river reach.

*Reply: The authors agree that the case study is partially engineered channel. However, it is not totally true that the study area contains a specific case. The study area is a typical case of small streams in South Korea. The tested cross-sections were once engineered or sometimes partially engineered. Then, nature has modified its shape as aging. The authors further include two more sites in the current study area to ensure covering different shapes of cross-sections. Furthermore, two additional simulation studies with U-shape and V-shape were made, accordingly. Hope this modification satisfactory to this reviewer.*

[Figure]

*Figure 10. Locations of four tested sites in the Migok-cheon stream. Note that the other four panels surrounding the left-top panel magnify each tested site by showing the point clouds of the observed data taken from the UAV photographs. The aerial images were taken from the authors.*

[Figure]

*Figure 13. Point cloud data (red circles) for Site-2 and model-fitted line (black dashed line) with KLR (panel(a)), LOWESS (panel(b)), and PolyFit (panel(c)) as well as the observed surveying. Note that (1) the observed line was drawn from the previous surveying in BRTMA (2019); and (2) the detailed information including the map is attached in Supplementary Material.*

[Figure]

*Figure 14. Point cloud data (red circles) for Site-3 and model-fitted line (black dashed line) with KLR (panel(a)), LOWESS (panel(b)), and PolyFit (panel(c)) as well as the observed surveying. Note that (1) the observed line was drawn from the previous surveying in BRTMA (2019); and (2) the detailed information including the map is attached in Supplementary Material.*

- The value of parameter "a" can not be set based on one idealized cross-section. The results show that the choice of "a" will change based on river characteristics and cannot be set to a particular value. As mentioned before, rivers exhibit a wide variety of shapes and they may also be a need to capture the riverine bedforms accurately. In such cases, the degree of smoothing will change based on river characteristics and therefore the value of "a" will change from case to case.

  *Reply: The authors appreciate this reviewer's insightful comment and agree that the "a" value might need to change in some cases. However, the authors consider that the "a" value is not much sensitive and we applied "a=2" to all three simulation and four sites without any problem. One can adopt this value to smooth further the cross-section. This has been further discussed in the discussion section. Hope this modification is acceptable to this reviewer.*

*"5. Discussion*
*Secondly, the KLR model should define the number of K-neighbors. The result of the tested model illustrates that the value of 1.5-2.5 for 'a' in Eq. (10) might be a good range. Further estimation procedure might be required in some cases to produce cross-sections that are more accurate. However, the value is not very sensitive at each case presented in the current study. Note that a=2.0 was employed -for the U-shape and V-shape synthetic cross-section and the case study without any further estimation procedure."*

  "

- Similar to the analysis for idealized cross-section, there should be a comparison of the differences between the cross-sections estimated using polynomial fitting and LOWESS. The differences should be analyzed not just using RMSE in elevation but also in area and perimeter curves for all regression techniques.

  *Reply: The authors appreciate this reviewer's detailed comment. Following this reviewer's comment, LOWESS and PolyFit models were applied to the case studies and compared. Further RMSE and MAE were included for the simulation study and the case study in Table 1 and Table 2, respectively. Hope this modification acceptable to this reviewer. Note that the area and perimeter part were removed from the other reviewer's comment.*

Some minor comments:

Given the overall drawback to the design of the study, I am not delving deeply into the minor corrections in the manuscript, but I have noted some of them.

- How do you decide on the in-stream variation in elevation?

  *Reply: The in-stream variation part (4.3.3) was removed to avoid confusion and the other reviewer's comment. Hope this modification acceptable to this reviewer. If this is not what this reviewer indicates, please further inform the authors in detail.*

- Define polyfit2 and polyfit4 in text.

  *Reply: PolyFit2,PolyFit3, and PolyFit4 were defined accordingly.*

*2.1 Polynomial Regression*

*A polynomial regression model can be used when the relationship between a predictor (x) and an explanatory variable (y) is nonlinear or curvilinear. The $M^{th}$-order polynomial regression can be expressed as*

$$y = \beta_0 + \beta_1 x + \beta_2 x^2 + \cdots + \beta_k x^M + \epsilon = \sum_{i=0}^{M} \beta_i x^i + \epsilon = \boldsymbol{x\beta} + \epsilon \qquad (1)$$

*where $\epsilon$ is considered to be a random noise with zero mean and M is the degree of the polynomial regression model, called PolyFit. Here, x can be the distance from the base location in a river cross-section with a length unit (meter, in the current study) and y is the elevation with the same length unit (meter as well).*

*According to its degree M, the model is structured as follows.*

$$y = \beta_0 + \beta_1 x + \beta_2 x^2 + \epsilon \qquad (2)$$
$$y = \beta_0 + \beta_1 x + \beta_2 x^2 + \beta_3 x^3 + \epsilon \qquad (3)$$
$$y = \beta_0 + \beta_1 x + \beta_2 x^2 + \beta_3 x^3 + \beta_4 x^4 + \epsilon \qquad (4)$$

*The models in Eq. (2), Eq. (3), and Eq. (4) are defined as PolyFit2, PolyFit3, and PolyFit4, respectively, and employed in the current study.*

- X-axis and Y-axis label and units missing in several of the figures

  *Reply: All the labels and units were added in the figures.*

[Figure]

*Figure 1. Assumed synthetic trapezoidal channel (not a real one) to test the KLR model (thick black dotted line) for the point cloud data with different portions of the number of neighbors ($k = a\sqrt{n}$, here a=1, 2, 3, and 4 at each panel). Note that (1) the trapezoidal sections are consistent with a 4 m top both sides and a 6 m base width as well as a 1:1 side slope with a 6 m height; (2) the number of points for the channel was divided at each 0.1 m to a total of 161 points (blue line); (3) 2 times the divided data are simulated with Eq.(17) to a total of 322 points (red dots); and (4) the elevation of the bottom channel was assumed to be 18 m.*

[Figure]

*Figure 2. Point cloud data (red circles) for Site-1 and model-fitted line (black dashed line) with KLR (panel(a)), LOWESS (panel(b)), and PolyFit (panel(c)) as well as the observed surveying. Note that (1) the observed line was drawn from the previous surveying in BRTMA (2019); and (2) the detailed information including the map is attached in Supplementary Material.*